# LEARNING DIFFUSION BRIDGES ON CONSTRAINED DOMAINS

**Xingchao Liu, Lemeng Wu, Mao Ye, Qiang Liu**
Department of Computer Science
University of Texas at Austin
{xcliu, lmwu, my21, lqiang}@cs.utexas.edu

## ABSTRACT

Diffusion models have achieved promising results on generative learning recently. However, because diffusion processes are most naturally applied on the unconstrained Euclidean space $\mathbb{R}^d$, key challenges arise for developing diffusion based models for learning data on constrained and structured domains. We present a simple and unified framework to achieve this that can be easily adopted to various types of domains, including product spaces of any type (be it bounded/unbounded, continuous/discrete, categorical/ordinal, or their mix). In our model, the diffusion process is driven by a drift force that is a sum of two terms: one singular force designed by *Doob's h-transform* that ensures all outcomes of the process to belong to the desirable domain, and one non-singular neural force field that is trained to make sure the outcome follows the data distribution statistically. Experiments show that our methods perform superbly on generating tabular data, images, semantic segments and 3D point clouds. Code is available at https://github.com/gnobitab/ConstrainedDiffusionBridge.

## 1 INTRODUCTION

Diffusion-based deep generative models, notably score matching with Langevin dynamics (SMLD) (Song & Ermon, 2019, 2020), denoising diffusion probabilistic models (DDPM) (Ho et al., 2020), and their variants (e.g., Song et al., 2020b,a; Kong & Ping, 2021; Song et al., 2021; Nichol & Dhariwal, 2021), have shown to achieve new state of the art results for image synthesis (Dhariwal & Nichol, 2021; Ramesh et al., 2022; Ho et al., 2022; Liu et al., 2021), audio synthesis (Chen et al., 2020; Kong et al., 2020), point cloud synthesis (Luo & Hu, 2021a,b; Zhou et al., 2021), and many other AI tasks. These methods train a deep neural network to drive as drift force a diffusion process to generate data, and are shown to outperform competitors, mainly GANs and VAEs, on stability and sample diversity (Xiao et al., 2021; Ho et al., 2020; Song et al., 2020b).

However, due to the continuous nature of diffusion processes, the standard approaches are restricted to generating unconstrained continuous data in $\mathbb{R}^d$. For generating data constrained on special structured domains, such as discrete, bound data or mixes of them, special techniques , e.g., dequantization (Uria et al., 2013; Ho et al., 2019) and multinomial diffusion (Hoogeboom et al., 2021; Austin et al., 2021), need to be developed case by case and the results still tend to be unsatisfying despite promising recent advances (Hoogeboom et al., 2021; Austin et al., 2021).

This work proposes a simple and unified framework for learning diffusion models on general constrained domains $\Omega$ embedded in the Euclidean space $\mathbb{R}^d$. The idea is to learn a continuous $\mathbb{R}^d$-valued diffusion process $Z_t$ on time interval $t \in [0, T]$, with a carefully designed force field, such that the final state $Z_T$ guarantees to 1) fall into the desirable domain $\Omega$, and 2) follows the data distribution asymptotically. We achieve both steps by leveraging a key tool in stochastic calculus called *Doob's h-transform* (Doob, 1984), which provides formula for deriving the diffusion processes whose final states are guaranteed to fall into a specific set or equal a specific value.

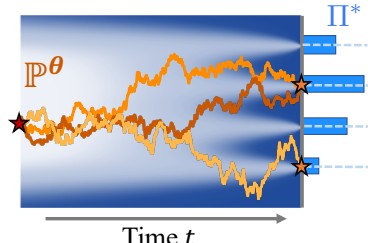

**Figure 1:** An $\Omega$-Bridge on discrete domain $\Omega = \{1, 2, 3, 4\}$.

---

**Algorithm 1** Learning Diffusion Models on Constrained Domains (a Simple Example)

---

**Input**: A dataset $\mathcal{D} := \{x^{(i)}\}$ drawn from distribution $\Pi^*$ on a domain $\Omega = \{e_1, e_2, \ldots, e_K\}$.

**Goal**: Learn a diffusion model that terminates at time $T$ to generate samples from $\Pi^*$.

**Learning**: Solve the optimization below with stochastic gradient descent (or other optimizers)

$$\theta^* = \arg\min_\theta \int_0^T \mathbb{E}_{x \sim \mathcal{D}} \left[ \left\| f^\theta(Z_t, t) - \nabla_{Z_t} \log \omega^\Omega(x \mid Z_t, t) \right\|^2 \right] \mathrm{d}t, \tag{1}$$

where

$$\omega^\Omega(x \mid z, t) = \frac{\exp\left(-\dfrac{\|z - x\|^2}{2(T - t)}\right)}{\displaystyle\sum_{e \in \Omega} \exp\left(-\dfrac{\|z - e\|^2}{2(T - t)}\right)}, \qquad Z_t = \frac{t}{T} x + (1 - \frac{t}{T}) x_0 + \sqrt{\frac{t(T - t)}{T}} \xi, \tag{2}$$

with $x$ drawn from the dataset $\mathcal{D}$, $\xi \sim \mathcal{N}(0, I)$, and $x_0$ any initial point.

**Sampling**: Generate sample $Z_T$ from

$$\mathrm{d}Z_t = \left[ f^{\theta^*}(Z_t, t) + \nabla_{Z_t} \log \sum_{e \in \Omega} \exp\left(-\frac{\|Z_t - e\|^2}{2(T - t)}\right) \right] \mathrm{d}t + \mathrm{d}W_t, \quad Z_0 = x_0.$$

**Remark**  When the domain $\Omega$ is a manifold (e.g., line or surface) in $\mathbb{R}^d$, simply replace the sum $\sum_{e \in \Omega}$ with the corresponding line or surface (in general Hausdorff) integration $\int_\Omega$ on $\Omega$.

---

Our simple procedure can be applied to any domain $\Omega$ once a properly defined summation (for discrete sets) or integration (for continuous domains) can be evaluated. To give a quick overview on the practical intuition without invoking the mathematical theory, we show in Algorithm 1 a simple instance of the framework when the domain is a discrete set $\Omega = \{e_1, \ldots, e_K\}$. The idea is to set up the diffusion model to have a form of

$$\mathrm{d}Z_t = \left[ f^\theta(Z_t, t) + \nabla_{Z_t} \psi^\Omega(Z_t, t) \right] \mathrm{d}t + \mathrm{d}W_t, \quad \psi^\Omega(z, t) := \log \sum_{e \in \Omega} \exp\left(-\frac{\|z - e\|^2}{2(T - t)}\right), \tag{3}$$

where the drift is a sum of a non-singular (e.g., bounded) term $f^\theta(z, t)$ which is a trainable neural force field with parameter $\theta$, and a singular term $\nabla_z \psi^\Omega(z, t)$, which drives $Z_t$ towards set $\Omega$ as a gradient ascent on $\psi^\Omega(z, t)$. The $\psi^\Omega(z, t)$ measures the closeness of $z$ to set $\Omega$, as the log-likelihood of a Gaussian mixture model (GMM) centered on the elements in $\Omega$ with variance $T - t$. When $t$ approaches to the terminal time $T$, the variance $T - t$ of the GMM goes to zero, and the magnitude of $\nabla_z \psi^\Omega(z, t)$ grows to infinity, hence ensuring that $Z_T$ must belong to $\Omega$. In particular, note that

$$\nabla_z \psi^\Omega(z, t) = \sum_{e \in \Omega} \omega^\Omega(e \mid z, t) \frac{e - z}{T - t}, \qquad \omega^\Omega(e \mid z, t) = \frac{\exp\left(-\frac{\|z - e\|^2}{2(T - t)}\right)}{\exp(\psi^\Omega(z, t))},$$

which increases with an $O(1/(T - t))$ rate as $t \to T$; here $\omega^\Omega(e \mid z, t)$ is the softmax probability measuring the relative closeness of $z$ to the elements $e$ in $\Omega$ (see also Eq (2)).

As we show in Section 2.3, once $f^\theta$ is non-singular in the sense of the mild condition of $\int_0^T \mathbb{E}[\|f^\theta(Z_t, t)\|^2]\mathrm{d}t < +\infty$, the diffusion model in (3) guarantees to yield a final state $Z_T$ that belongs to $\Omega$, and hence provides a flexible model family on $\Omega$. Moreover, as shown in Eq 1 in Algorithm 1, the neural field $f^\theta$ can be simply trained to approximate $\nabla \log \omega^\Omega(e \mid z, t)$ with $e$ plugged as the data point that we expect to achieve when starting from $z$ at time $t$. Intuitively, such fitted $f^\theta$ increases the relative probability of the observed data points and hence allows us to fit the data distribution. Empirically, diffusion models learned through $\Omega$-bridge achieves favorable results in generating mixed discrete/continuous tabular data, point clouds on grids, categorical semantic segments and discrete CIFAR10 images.

**Outline** The rest of the paper is organized as follows. Section 2 introduces $h$-transform, which allows us to derive bridge processes that are guaranteed to enter specific sets at the terminal time, and Section 2.3 specifies the parametric diffusion models for $\Omega$-bridges. Then, with the learnable diffusion models, Section 3 introduces the general learning framework along with the loss function.

## 2 BACKGROUND: DIFFUSION PROCESSES AND $h$-TRANSFORM

A diffusion process $\boldsymbol{Z} = \{Z_t \colon t \in [0, T]\}$ on $\mathbb{R}^d$ follows a stochastic differential equation of form

$$\mathbb{Q}: \qquad \mathrm{d}Z_t = b(Z_t, t)\mathrm{d}t + \sigma(Z_t, t)\mathrm{d}W_t, \tag{4}$$

where $W_t$ is a Wiener process, and $\sigma \colon [0, T] \times \mathbb{R}^d \to \mathbb{R}$ is a positive diffusion coefficient, and $b \in [0, T] \times \mathbb{R}^d \to \mathbb{R}^d$ is a drift function. We use $\mathbb{Q}$ (or $\mathbb{P}$) to denote the path measure of stochastic processes $\boldsymbol{Z}$, which are probability measures on the space of continuous paths. Let $\mathbb{Q}_t$ be the marginal distribution of $Z_t$ at time $t$ under $\mathbb{Q}$.

Our framework heavily relies on the *bridge processes*, special stochastic processes that guarantee to achieve a deterministic value or fall into a given set at the final state $T$.

☞ *For a set $\Omega \subseteq \mathbb{R}^d$, a process $Z$ in $\mathbb{R}^d$ with law $\mathbb{Q}$ is called an $\Omega$-bridge if $\mathbb{Q}(Z_T \in \Omega) = 1$.*

One natural approach to constructing bridge processes is to derive the conditioned process of a general unconstrained process given that the desirable bridge constraint happens. Specifically, assume that $\mathbb{Q}$ is the law of a general unconstrained diffusion process of form (4), and denote by $\mathbb{Q}^\Omega(\cdot) = \mathbb{Q}(\cdot \mid Z_T \in \Omega)$ the conditioned distribution given that the event of $Z_T \in \Omega$ happens. Then $\mathbb{Q}^\Omega$ is guaranteed to be an $\Omega$-bridge by definition. Importantly, a remarkable result from Doob (Doob, 1984), now known as $h$-transform, shows that $\mathbb{Q}^\Omega$ is the law of a diffusion process with a properly modified drift term. Below, we introduce this results, first for the case $x$-bridge when $\Omega = \{x\}$ includes a single point, and then for more general sets $\Omega$. For simplicity, we only state the formula from $h$-transform that are useful for us without proofs. See e.g., Oksendal (2013); Rogers & Williams (2000) for more background on $h$-transform.

### 2.1 $x$-BRIDGES

Let us first consider the $x$-bridge $\mathbb{Q}^x(\cdot) \coloneqq \mathbb{Q}(\cdot \mid Z_T = x)$, the process $\mathbb{Q}$ pinned at a deterministic terminal point $Z_T = x$. By the result from $h$-transform (see e.g., Oksendal (2013)), the conditioned process $\mathbb{Q}^x(\cdot) \coloneqq \mathbb{Q}(\cdot \mid Z_T = x)$, if it exists, can be shown to be the law of

$$\mathrm{d}Z_t = \left(b(Z_t, t) + \sigma^2(Z_t, t)\nabla_z \log q_{T|t}(x \mid Z_t)\right)\mathrm{d}t + \sigma(Z_t, t)\mathrm{d}W_t, \tag{5}$$

where $q_{T|t}(x|z)$ is the density function of the transition probability

$$\mathbb{Q}_{T|t}(\mathrm{d}x \mid z) \coloneqq \mathbb{Q}(Z_T \in \mathrm{d}x \mid Z_t = z),$$

where $\mathrm{d}x$ denotes an infinitesimal volume centering around $x$. Compared with the diffusion process (4) of $\mathbb{Q}$, the main difference is that the conditioned process has an additional drift force $\sigma^2(z, t)\nabla_z \log q_{T|t}(x|z)$ which plays the role of steering $Z_t$ towards the target $Z_T = x$; this is a singular force whose magnitude increases to infinity as $t \to T$, because $q_{T|t}(\cdot \mid z)$ is a delta measure centered at $z$ when $t = T$.

In addition, by Bayes rule, the distribution of the initial state $Z_0$ should be given by

$$Z_0 \sim \mathbb{Q}_{0|T}(\cdot \mid x), \qquad\qquad \mathbb{Q}_{0|T}(\mathrm{d}z|x) \propto \mathbb{Q}_0(\mathrm{d}z)q_{T|0}(x|z). \tag{6}$$

**Example 2.1.** *If $\mathbb{Q}$ is the law of $\mathrm{d}Z_t = \sigma_t\mathrm{d}W_t$, we have $\mathbb{Q}_{T|t}(\cdot|z) = \mathcal{N}(z, \beta_T - \beta_t)$, where $\beta_t = \int_0^t \sigma_s^2\mathrm{d}s$. Hence, following the formula in (5), $\mathbb{Q}^x \coloneqq \mathbb{Q}(\cdot|Z_T = x)$ is the law of*

$$\mathrm{d}Z_t = \sigma_t^2 \frac{x - Z_t}{\beta_T - \beta_t}\mathrm{d}t + \sigma_t\mathrm{d}W_t, \tag{7}$$

*and $Z_0 \sim \mathbb{Q}_{0|T}(\mathrm{d}z) \propto \mathbb{Q}_0(\mathrm{d}z)\phi(x \mid z, \beta_T - \beta_t)$, and $\phi(\cdot|\mu, \sigma^2)$ is the density function of $\mathcal{N}(\mu, \sigma^2)$. The process in (7) is known as a (time-scaled) Brownian bridge. Note that the drift in (7) grows to infinity in magnitude with a rate of $O(1/(\beta_T - \beta_t))$ as $t \to T$, which ensures that $Z_t = x$ with probability one.*

**Arbitrary initialization** To make (5) the conditioned process of (4), the initial distribution must follow the Bayes rule in (6). However, thanks to the singular force $\nabla_z \log q_{T|t}(x|z)$, the process (5) can guarantee $Z_t^x = x$ from an arbitrary initialization once the process is well defined. When the initialization is different from (6), the process in (5) is no longer the conditioned process of (4), but it remains to be an $x$-bridge in that $Z_T = x$ is still guaranteed. To see why this is the case, assume that $\mathbb{Q}$ is initialized from a deterministic point $Z_0 = x_0$. Then we would still have $Z_0 = x_0$ when conditioned on $Z_T \in \Omega$ by Bayes rule. This suggests that (5) starting from any deterministic initialization is the condition process of $\mathbb{Q}$ with the same deterministic initialization, and is hence an $x$-bridge. As a result, (5) from any stochastic initialization is also an $x$-bridge because it can be viewed as the mixture of the processes with different deterministic initialization, all of which are $x$-bridges. See Appendix A.4 for a detailed analysis, in which it is shown that (5) with an arbitrary initialization can be viewed as the conditioned process of a special class of non-Markov processes called *reciprocal process*.

## 2.2 $\Omega$-BRIDGES

More generally, for the law $\mathbb{Q}$ of (4) and a set $\Omega \in \mathbb{R}^d$, the $\Omega$-bridge $\mathbb{Q}^\Omega := \mathbb{Q}(\cdot \mid Z_T \in \Omega)$ follows

$$\mathbb{Q}^\Omega : \qquad \mathrm{d}Z_t = \eta^\Omega(Z_t, t)\mathrm{d}t + \sigma(Z_t, t)\mathrm{d}W_t, \tag{8}$$

with $\quad \eta^\Omega(z, t) = b(z, t) + \sigma^2(z, t)\mathbb{E}_{x \sim \mathbb{Q}_{T|t,z,\Omega}}[\nabla_z \log q_{T|t}(x \mid z)], \quad Z_0 \sim \mathbb{Q}_{0|T}(\cdot \mid Z_T \in \Omega),$

where drift force $\eta^\Omega$ is similar to that of the $x$-bridge in (5), except that the final state $x$ is now randomly drawn from an $\Omega$-truncated (or $\Omega$-conditioned) transition probability:

$$\mathbb{Q}_{T|t,z,\Omega}(\mathrm{d}x \mid z) := \mathbb{Q}(Z_T \in \mathrm{d}x \mid Z_t = z, Z_T \in \Omega),$$

which is the transition probability from $Z_t$ to $Z_T$, conditioned on that $Z_T \in \Omega$. In practice, its form can be derived using Bayes rule.

**Example 2.2.** *Assume $\mathbb{Q}$ follows $\mathrm{d}Z_t = \sigma_t \mathrm{d}W_t$. Then $\mathbb{Q}^\Omega$ yields the following $\Omega$-bridge:*

$$\mathrm{d}Z_t = \eta^\Omega(Z_t, t)\mathrm{d}t + \sigma_t \mathrm{d}W_t, \qquad \eta^\Omega(z, t) = \sigma_t^2 \mathbb{E}_{x \sim \mathcal{N}_\Omega(z, \beta_T - \beta_t)}\left[\frac{x - z}{\beta_T - \beta_t}\right], \tag{9}$$

*where $\mathcal{N}_\Omega(z, \sigma^2) = \mathrm{Law}(Z \mid Z \in \Omega)$ with $Z \sim \mathcal{N}(\mu, \sigma^2)$, which is an $\Omega$-truncated Gaussian distribution $\mathcal{N}(\mu, \sigma^2)$, whose density function is $\phi_\Omega(x) \propto \mathbb{I}(x \in \Omega)\phi(x|\mu, \sigma)$ with $\phi(x|\mu, \sigma)$ the density function of $\mathcal{N}(\mu, \sigma^2)$.*

*Note that it is tractable to calculate $\eta^\Omega$ once we can evaluate the expectation of $\mathcal{N}_\Omega(z, \beta_T - \beta_t)$. A general case is when $\Omega = I_1 \times \cdots I_d$, for which the expectation reduces to one dimensional Gaussian integrals. See Appendix A.6 and A.7 for details and examples of $\eta^\Omega$.*

As in the $x$-bridge, we can set the initialization to be any distribution supported on the set of points that can reach $\Omega$ following $\mathbb{Q}$ (precisely, points $z_0$ that satisfy $\Omega \cap \mathrm{supp}(\mathbb{Q}_T(\cdot|Z_0 = z_0)) \neq \emptyset$) using the mixture of initialization argument.

## 2.3 A PARAMETRIC FAMILY OF $\Omega$-BRIDGES

The formula in (8) only provides a fixed process for a given $\mathbb{Q}$. For the purpose of learning generative models, however, we need a rich family of $\Omega$-bridges within which we can search for a best one to fit with the data distribution. It turns out we can achieve this by simply adding an extra non-singular drift force, which can be a trainable neural network, on top of the $\Omega$-bridge in (8). Specifically, we construct the following parametric diffusion model $\mathbb{P}^\theta$:

$$\mathbb{P}^\theta : \qquad \mathrm{d}Z_t = (\sigma(Z_t, t)f^\theta(Z_t, t) + \eta^\Omega(Z_t, t))\mathrm{d}t + \sigma(Z_t, t)\mathrm{d}W_t, \quad Z_0 \sim \mathbb{P}_0^\theta, \tag{10}$$

where $f^\theta(z, t)$ is a neural network with input $(z, t)$ and parameter $\theta$, which will be trained based on the empirical observations. Adding the neural drift $\sigma(Z_t, t)f^\theta(Z_t, t)$ term does not break the $\Omega$-bridge condition, once it satisfies a very mild regularization condition:

**Proposition 2.3.** *For any $\mathbb{Q}^\Omega$ following $\mathrm{d}Z_t = \eta^\Omega(Z_t, t)\mathrm{d}t + \sigma(Z_t, t)\mathrm{d}W_t$ that is an $\Omega$-bridge, the $\mathbb{P}^\theta$ in (10) is also an $\Omega$-bridge if $\mathbb{E}_{Z \sim \mathbb{Q}^\Omega}[\int_0^T \|f^\theta(Z_t, t)\|_2^2 \mathrm{d}t] < +\infty$ and $\mathcal{KL}(\mathbb{Q}_0^\Omega \| \mathbb{P}_0^\theta) < +\infty$.*

The condition on $f^\theta$ is very mild, and it is satisfied if $f^\theta$ is bounded. Moreover, it can easily hold even when $f_\theta$ is not bounded. For example, assuming that $f_\theta(x) \leq a \|x\|^\beta + b$, which holds for ReLU network with $\beta = 1$, we just need to require that the underlying process has a bounded moment $E_{Z \sim Q^\Omega} [\int_0^T \|Z_t\|^{2\beta} \, dt] < +\infty$, which is a typical regularity condition to expect.

## 3 LEARNING $\Omega$-BRIDGE MODELS

Let $\{x^{(i)}\}_{i=1}^n$ be an i.i.d. sample from an unknown distribution $\Pi^*$ on a domain $\Omega \subseteq \mathbb{R}^d$. Our goal is to learn the parameter $\theta$ for the $\Omega$-bridge model $\mathbb{P}^\theta$ in (10) such that the terminal distribution $Z_T \sim \mathbb{P}_T^\theta$ matches the data $X \sim \Pi^*$. We should distinguish $\mathbb{P}^\theta$, which is the trainable generative model, and $\mathbb{Q}$, which is a fixed "baseline process" that helps us to derive methods for constructing and learning the model. $\mathbb{Q}$ can be the simple Brownian motion in Example 2.1 and 2.2.

As the case of other diffusion models, $\mathbb{P}^\theta$ can be viewed as a model with an infinite dimensional latent variable of the intermediate trajectories of $\mathbf{Z}$. Hence, a canonical learning approach is expectation maximization (EM), which alternates between

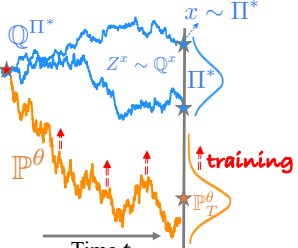

1) E-step: estimating the posterior $\mathbb{P}^{\theta,x} := \mathbb{P}^\theta(\mathbf{Z} \mid Z_T = x)$ of the latent trajectories $\mathbf{Z}$ given the observation $Z_T = x$;

2) M-step: estimating the parameter $\theta$ with $\mathbf{Z}$ imputed from $\mathbb{P}^{\theta,x}$.

A key challenge, however, is that the posterior distribution $\mathbb{P}^{\theta,x}$ is difficult to calculate due to the presence of neural force field in $\mathbb{P}^\theta$ (as the $h$-transform formula would have no closed form), and it need to be iteratively updated as $\theta$ changes. Following DDPM (Ho et al., 2020), we consider a simpler approach that replaces the posterior $\mathbb{P}^{\theta,x}$ with an arbitrary $x$-bridge, denoted by $\mathbb{Q}^x$. This yields a simplified EM algorithm without the expensive posterior inference in the E-step. A natural choice is the conditioned process $\mathbb{Q}^x := \mathbb{Q}(\mathbf{Z} \mid Z_T = x)$, but the method works for a general $x$-bridge.

Specifically, let $\mathbb{Q}^{\Pi^*}(\cdot) = \int \mathbb{Q}^x(\cdot)\Pi^*(\mathrm{d}x)$ be the mixture of the $x$-bridges whose end point $x$ is randomly drawn from the data distribution $x \sim \Pi^*$. The trajectories from $\mathbb{Q}^{\Pi^*}$ can be generated in the following "backward" way: first drawing a data point $x \sim \Pi^*$, and then $\mathbf{Z} \sim \mathbb{Q}^x$ conditioned on the end point $x$. Obviously, by construction, the terminal distribution of $\mathbb{Q}^{\Pi^*}$ equals $\Pi^*$, that is, $\mathbb{Q}_T^{\Pi^*} = \Pi^*$. Then, the model $\mathbb{P}^\theta$ can be estimated by fitting data drawn from $\mathbb{Q}^{\Pi^*}$ using maximum likelihood estimation:

$$\min_\theta \left\{ \mathcal{L}(\theta) := \mathcal{KL}(\mathbb{Q}^{\Pi^*} \| \mathbb{P}^\theta) \right\}. \tag{11}$$

The classical (variational) EM would alternatively update $\theta$ (M-step) and $\mathbb{Q}^x$ (E-step) to make $\mathbb{Q}^x \approx \mathbb{P}^{\theta,x}$. Why is it OK to simply drop the E-step? At the high level, it is the benefit from using universal approximators like deep neural networks: if the model space of $\mathbb{P}^\theta$ is sufficiently rich, by minimizing the KL divergence in (11), $\mathbb{P}^\theta$ can approximate the given $\mathbb{Q}^{\Pi^*}$ well enough (in a way that is made precise in the Appendix A) such that their terminal distributions are close: $\mathbb{P}_T^\theta \approx \mathbb{Q}_T^{\Pi^*} = \Pi^*$.

☞ *Learning latent variable models require no E-step if the model space is sufficiently rich.*

We should see that in this case the latent variables $Z$ in the learned model $\mathbb{P}^\theta$ is *dictated* by the choice of the imputation distribution $\mathbb{Q}$ since we have $\mathbb{P}^{\theta,x} = \mathbb{Q}^x$ when the KL divergence in (11) is fully minimized to zero; EM also achieves $\mathbb{P}^{\theta,x} = \mathbb{Q}^x$ but has the imputation distribution $\mathbb{Q}^x$ determined by the model $\mathbb{P}^\theta$, not the other way.

**Loss Function** In its general form, the $x$-bridge $\mathbb{Q}^x$ that we use can be a non-Markov diffusion process

$$\mathbb{Q}^x: \quad \mathrm{d}Z_t = \eta^x(\mathbf{Z}, t)\mathrm{d}t + \sigma(Z_t, t)\mathrm{d}W_t, \quad Z_0 \sim \mu^x, \tag{12}$$

which has the same diffusion coefficient $\sigma(Z_t, t)$ as $\mathbb{P}^\theta$ in (10), and any $x$-dependent $\eta^x$ and initialization $\mu^x$ once the $x$-bridge condition is ensured. As the general framework, we assume that $\eta^x$ can depend on the whole trajectory $\mathbf{Z}$.

---

**Algorithm 2** Learning $\Omega$-Bridge Diffusion Models

---

**Input**: A dataset $\mathcal{D} := \{x^{(i)}\}$ drawn from distribution $\Pi^*$ on a domain $\Omega$.
**Setup**: Specify an $x$-bridge $\mathbb{Q}^x$ and an $\Omega$-bridge $\mathbb{Q}^\Omega$

$$\mathbb{Q}^x: \ \mathrm{d}Z_t = \eta^x(\boldsymbol{Z}, t)\mathrm{d}t + \sigma(Z_t, t)\mathrm{d}W_t, \qquad \mathbb{Q}^\Omega: \ \mathrm{d}Z_t = \eta^\Omega(Z_t, t)\mathrm{d}t + \sigma(Z_t, t)\mathrm{d}W_t,$$

Specify the generative model $\mathbb{P}^\theta$ based on $\mathbb{Q}^\Omega$ and a neural network $f^\theta$:

$$\mathbb{P}^\theta: \ \ \mathrm{d}Z_t = (\sigma(Z_t, t)f^\theta(Z_t, t) + \eta^\Omega(Z_t, t))\mathrm{d}t + \sigma(Z_t, t)\mathrm{d}W_t, \quad Z_0 \sim \mathbb{P}_0^\theta.$$

*Default: let $\mathbb{Q}$ be the law of $\mathrm{d}Z_t = \sigma_t\mathrm{d}W_t$ and derive the bridges by h-transform as $\mathbb{Q}^x = \mathbb{Q}(\cdot|Z_T = x)$ in Eq (7) and $\mathbb{Q}^\Omega = \mathbb{Q}(\cdot|Z_T \in \Omega)$ in Eq (9).*
**Training**: Estimating $\theta$ by minimizing the loss function (13) using any off-the-shelf optimizer.
**Sampling**: Generate sample $Z_T$ from $\mathbb{P}^\theta$ with the trained parameter $\theta$.

---

Using Girsanov theorem (e.g., Oksendal, 2013), with $\mathbb{P}^\theta$ in (10) and $\mathbb{Q}^x$ in (12), the KL divergence in (11) can be shown to equal to

$$\mathcal{L}(\theta) = \mathbb{E}_{\substack{x \sim \Pi^* \\ Z \sim \mathbb{Q}^x}}\left[\underbrace{-\log p_0^\theta(Z_0)}_{\text{MLE of initial dist.}} + \frac{1}{2}\int_0^T \underbrace{\left\|\sigma^{-1}(Z_t, t)(s^\theta(Z_t, t) - \eta^x(\boldsymbol{Z}, t))\right\|^2}_{\text{score matching}}\mathrm{d}t\right] + const, \quad (13)$$

where we write $s^\theta$ as the overall drift force of $\mathbb{P}^\theta$ in (10), that is,

$$s^\theta(z, t) = \sigma(z, t)f^\theta(z, t) + \eta^\Omega(z, t),$$

and $p_0^\theta$ is the probability density function (PDF) of the initial distribution $\mathbb{P}_0^\theta$. Therefore, $\mathcal{L}(\theta)$ is a sum of the negative log-likelihood of the initial distribution that encourages $\mathbb{P}_0^\theta \approx \mathbb{Q}_0^{\Pi^*}$, and a least squares loss between $s^\theta$ and $\eta^x$. In practice, we simply fix the initial distribution $\mathbb{P}_0^\theta$ to be a delta measure on a fixed point (say $x_0 = 0$), so we only need to train the drift function $f^\theta$. Algorithm 1 shows a simple instance of the framework when the baseline process $\mathbb{Q}$ is the standard Brownian motion $\mathrm{d}Z_t = \mathrm{d}W_t$, $\mathbb{Q}^x = \mathbb{Q}(\cdot \mid Z_T = x)$ and $\sigma(z, t) = 1$. Note that the least squares term in (13) can be viewed as enforcing $f^\theta \approx \sigma^{-1}(\eta^x - \eta^\Omega)$, which reduces to $f^\theta \approx \nabla \log \omega^\Omega$ in the case of Algorithm 1.

**Related Works**  Bridge processes provide a simple and flexible approach to learning diffusion generative models, which was explored in Peluchetti (2021); Ye et al. (2022); Wu et al. (2022); De Bortoli et al. (2021). Heng et al. (2021) investigates the orthogonal problem of simulating from the bridge $\mathbb{Q}^x$ for a given $\mathbb{Q}$. In comparison, our method learns diffusion models on general domains $\Omega$ on which an $\Omega$-bridge can be derived (using $h$-transform or any other method), and hence provides a highly flexible framework for learning with structured data (including discrete, continuous, and their mixes). This distinguishes it with existing approaches that are designed for special types of data (e.g., Ho et al. (2020); Hoogeboom et al. (2021); Austin et al. (2021); Li et al. (2022); Dieleman et al. (2022) for discrete data). De Bortoli et al. (2022) discusses how to learn score-based generative models on general Riemannian manifolds. Another highly related work is Ye et al. (2022), which proposes to learn first hitting diffusion models for generating data on both discrete sets and spheres. The advantage of our approach is that it is simpler and easier to derive for more complex types of domains.

## 4    EXPERIMENTS

We evaluate our algorithms for generating mixed-typed tabular data, grid-valued point clouds, categorical semantic segmentation maps, discrete CIFAR10 images. We observe that $\Omega$-bridge provides a particularly attractive and superb approach to generating data from various constrained domains.

**Algorithm Overview**    For all experiments, we use Algorithm 2 with the default choice of $\mathbb{Q}^x$ in (7) and $\mathbb{Q}^\Omega$ in (9). The specific form of $\eta^\Omega$ is derived based on the specific choice of the domain $\Omega$. By default, we set the initialization $Z_0 = 0$ and the optimizer Adam.

|  | Logistic ($\uparrow$) | AdaBoost ($\uparrow$) | MLP ($\uparrow$) |
|---|---|---|---|
| Real Training Data | 0.877±0.021 | 0.912±0.013 | 0.897±0.012 |
| TVAE (Xu et al., 2019) | 0.825±0.012 | 0.876±0.005 | 0.845±0.008 |
| CTGAN (Xu et al., 2019) | 0.649±0.014 | 0.841±0.021 | 0.843±0.016 |
| CopulaGAN (Patki et al., 2016) | 0.683±0.015 | 0.859±0.004 | 0.853±0.009 |
| Mixed-Bridge | **0.868±0.010** | **0.884±0.005** | **0.877±0.006** |

**Table 1:** Classification accuracy on the Adult Income dataset with different classifiers when trained with data synthesized by generative models. Real Training Data shows the upper bound of the metrics.

### 4.1 GENERATING MIXED-TYPE TABULAR DATA

Learning to generate tabular data is challenging, because tabular data usually contains a mixture of discrete and continuous attributes (Xu et al., 2019; Park et al., 2018). Unlike carefully designing special GANs as in previous works (Xu et al., 2019; Srivastava et al., 2017), $\Omega$-bridge can be seamlessly applied to mixed-typed tabular data generation without any further modification. In contrast, diffusion processes that solely work on discrete domain (Austin et al., 2021; Hoogeboom et al., 2021) cannot be applied to this task.

In this experiment, we use the *Adult Income* dataset (Kohavi, 1996), which contains 30,162 training samples. The data points are described by a series of attributes, including continuous (`age`, `capital-gain`, etc.) and discrete (`sex`, `race`, etc.). We compare with conditional tabular GAN (Xu et al., 2019) (CTGAN), CopulaGAN (Patki et al., 2016), and Table VAE (Xu et al., 2019) (TVAE), which are state-of-the-art GAN-based and VAE-based generative models for mixed-typed tabular data. Following previous works (Xu et al., 2019; Patki et al., 2016), we measure the classification accuracy on the real data of logistic regression, AdaBoost classifier and MLP classifier when trained on the generated data.

In the $\Omega$-bridge model, we set $\sigma_t = 3\exp(-3t)$ and $f^\theta$ a 3-layer MLP. In this case, $\Omega = I_1 \times \cdots \times I_{15}$, where $I_1$ to $I_9$ are discrete domains and $I_{10}$ to $I_{15}$ are non-negative continuous domains. For discrete domains, $I = \{e_1, \ldots, e_d\}$, we have, $\eta^I(z, t) = \sigma_t^2 \nabla_z \log \sum_{e \in I} \exp\left(-\frac{\|z - e\|^2}{2(\beta_T - \beta_t)}\right)$; for non-negative continuous domains, $I = [0, +\infty)$, derivation shows $\eta^I(z, t) = \sigma_t^2 \nabla_z \log\left(F(\frac{z}{\sqrt{\beta_T - \beta_t}})\right)$, where $F$ is the standard Gaussian CDF. Finally, we have $\eta^\Omega(z, t) = \sum_{i=1}^{15} \eta^{I_i}(z_i, t)$ for the whole domain $\Omega$. We set the number of diffusion steps to $K = 2000$. Results are shown in Table 1.

**Result** All the three different classifiers yield the highest accuracy when trained on the data generated by our method, referred to as *Mixed-Bridge* in this case. The result reflects that the data generated by Mixed-Bridge is closer to the real distribution than the baseline methods.

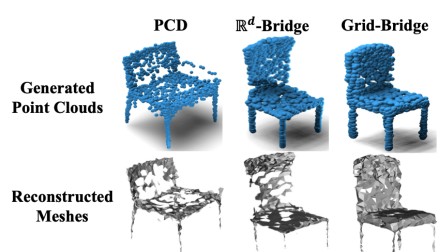

| Method | MMD $\downarrow$ | COV $\uparrow$ | 1-NNA $\downarrow$ |
|---|---|---|---|
| PCD (Luo & Hu, 2021a) | 13.37 | 46.60 | 58.94 |
| $\mathbb{R}^d$-Bridge | 13.30 | 46.52 | 59.32 |
| Grid-Bridge | **12.85** | **47.78** | **56.25** |

**Figure 2 & Table 2:** The point clouds (upper row) generated by different methods and meshes reconstructed from them (lower row). Grid-Bridge obtains more uniform points and hence better mesh thanks to the integer constraints. Numbers in the table are multiplied by $10^3$.

### 4.2 GENERATING INTEGER-VALUED POINT CLOUDS

A feature of point clouds in 3D objects in graphics is that they tend to distribute evenly, especially if they are discretized from a mesh. This aspect is omitted in most existing works on point cloud generation. As a result they tend to generate non-uniform points that are unsuitable for real applications, which often involve converting back to meshes with procedures like Ball-Pivoting (Bernardini

**Figure 3:** Results on generating categorical segmentation maps. Each pixel here an `one-hot` vector. Each dimension of the $\Omega$-bridge starts from a deterministic and evolve through a stochastic trajectory to converge to either 0 or 1. The generated samples have similar visual quality to the training data.

| Methods | ELBO ($\downarrow$) | IWBO ($\downarrow$) |
|---|---|---|
| Uniform Dequantization (Uria et al., 2013) | 1.010 | 0.930 |
| Variational Dequantization (Ho et al., 2019) | 0.334 | 0.315 |
| Argmax Flow (Softplus thres.) (Hoogeboom et al., 2021) | 0.303 | 0.290 |
| Argmax Flow (Gumbel distr.) (Hoogeboom et al., 2021) | 0.365 | 0.341 |
| Argmax Flow (Gumbel thres.) (Hoogeboom et al., 2021) | 0.307 | 0.287 |
| Multinomial Diffusion (Hoogeboom et al., 2021) | 0.305 | - |
| Cat.-Bridge (Constant Noise) | 0.844 | 0.707 |
| Cat.-Bridge (Noise Decay A) | **0.276** | **0.232** |
| Cat.-Bridge (Noise Decay B) | 0.301 | 0.285 |
| Cat.-Bridge (Noise Decay C) | 0.363 | 0.302 |

**Table 3:** Results on the CityScapes dataset. Cat. refers to 'Categorical'.

et al., 1999). We apply our method to generate point clouds that constrained on a integer grid which we show yields much more uniformly distributed points. To the best of our knowledge, we are the first work on integer-valued 3D point cloud generation.

A point cloud is a set of points $\{x_i\}_{i=1}^m$, $x_i \in \mathbb{R}^3$ in the 3D space, where $m$ refers to the number of points. We apply two variants of our method: $\mathbb{R}^d$-Bridge and Grid-Bridge. $\mathbb{R}^d$-Bridge generates points in the continuous 3D space, i.e., $\Omega_{\mathbb{R}} = \mathbb{R}^{3m}$. Grid-Bridge generate points that on integer grids, $\Omega_{\text{Grid}} = \{1, \ldots, 128\}^{3m}$. We fix the diffusion coefficient $\sigma_t = 1$. The number of diffusion steps $K$ is set to 1000. We test our method on ShapeNet (Chang et al., 2015) chair models, and compare it with Point Cloud Diffusion (PCD) (Luo & Hu, 2021a), a state-of-the-art continuous diffusion-based generative model for point clouds. The neural network $f^\theta$ in our methods are the same as that of PCD for fair comparison. Qualitative results and quantitative results are shown in Figure 2 and Table 2. As common practice (Luo & Hu, 2021a,b), we measure minimum matching distance (MMD), coverage score (COV) and 1-NN accuracy (1-NNA) using Chamfer Distance (CD) with the test dataset.

**Result**   Both $\mathbb{R}^d$-Bridge and Grid-Bridge get better MMD, COV, and 1-NNA than PCD. Moreover, by constraining the domain of interest to the integer grids, Grid-Bridge yields even better performance than $\mathbb{R}^d$-Bridge. In Figure 2, since the point clouds generated by Grid-Bridge are limited to integer grids, the reconstructed meshes from Ball-Pivoting clearly have higher quality than $\mathbb{R}^d$-Bridge and PCD.

## 4.3   GENERATING SEMANTIC SEGMENTATION MAPS ON CITYSCAPES

We consider unconditionally generating categorical semantic segmentation maps. We represent each pixels as a `one-hot` categorical vector. Hence the data domain is $\Omega = \{e_1, \ldots, e_c\}^{h \times w}$, where $c$ is the number of classes and $e_i$ is the $i$-th $c$-dimensional one-hot vector, and $h, w$ represent the height and width of the image. In CityScapes (Cordts et al., 2016), $h = 32, w = 64, c = 8$. In this experiment, we test different schedule of the diffusion coefficient $\sigma_t$, including *(Constant Noise)*: $\sigma_t = 1$; *(Noise Decay A)*: $\sigma_t = a\exp(-bt)$; *(Noise Decay B)*: $\sigma_t = a(1-t)$; *(Noise Decay C)* $\sigma_t = a(1 - \exp(-b(1-t)))$. Here $a$ and $b$ are hyper-parameters. The number of diffusion steps $K$ is set to 500. We measure the negative log-likelihood (NLL) of the test set using the learned models. The NLL (bits-per-dimension) is estimated with evidence lower bound (ELBO) and importance weighted bound (IWBO) (Burda et al., 2016), respectively, as in (Hoogeboom et al., 2021). We compare $\Omega$-Bridge with a state-of-the-art categorical diffusion algorithm, Argmax Flow (and Multinomial Diffusion) (Hoogeboom et al., 2021), and the traditional methods, uniform dequantization (Uria

| Methods | IS ($\uparrow$) | FID ($\downarrow$) | NLL ($\downarrow$) |
|---|---|---|---|
| **Discrete** | | | |
| D3PM uniform $L_{vb}$ (Austin et al., 2021) | 5.99 | 51.27 | 5.08 |
| D3PM absorbing $L_{vb}$ (Austin et al., 2021) | 6.26 | 41.28 | 4.83 |
| D3PM Gauss $L_{vb}$ (Austin et al., 2021) | 7.75 | 15.30 | 3.966 |
| D3PM Gauss $L_{\lambda=0.001}$ (Austin et al., 2021) | 8.54 | 8.34 | 3.975 |
| D3PM Gauss + logistic $L_{\lambda=0.001}$ | 8.56 | 7.34 | 3.435 |
| Integer-Bridge (Init. A) | **8.77** | **6.77** | 3.46 |
| Integer-Bridge (Init. B) | 8.68 | 6.91 | **3.35** |
| Integer-Bridge (Init. C) | 8.72 | 6.94 | 3.40 |

**Table 4:** Discrete CIFAR10 Image Generation

et al., 2013) and variational dequantization (Ho et al., 2019). The numerical results of the baselines are directly adopted from (Hoogeboom et al., 2021), and experiment configuration is kept the same for fair comparison. The neural network $f^\theta$ is the same as (Hoogeboom et al., 2021). The results are shown in Figure 3 and Table 3. Our $\Omega$-bridge is named *Categorical-Bridge (Cat.-Bridge)* in this experiment.

**Result** We observe that all the four kinds of Cat.-Bridge can successfully generate categorical semantic segments, and different noise schedules result in different empirical performance. Among the four variants of Cat.-Bridge, Cat.-Bridge with Noise Decay A yields the best ELBO and IWBO, surpassing all the other algorithms in comparison.

### 4.4 GENERATING DISCRETE CIFAR10 IMAGES

In this experiment, we apply three types of bridges. All of these bridges use the same output domain $\Omega = \{0, \ldots, 255\}^{h \times w \times c}$, where $h, w, c$ are the height, width and number of channels of the images, respectively. We set $\sigma_t = 3\exp(-3t)$. We consider different initial distributions: *(Init. A)* $Z_0 = 128$; *(Init. B)* $Z_0 = \hat{\mu}_0$, *(Init. C)* $Z_0 \sim \mathcal{N}(\hat{\mu}_0, \hat{\sigma}_0)$, where $\hat{\mu}_0$ and $\hat{\sigma}_0$ are the empirical mean and variance of pixels in the CIFAR10 training set. The number of diffusion steps $K$ is set to 1000. We compare with the variants of a state-of-the-art discrete diffusion model, D3PM (Austin et al., 2021). For fair comparison, we use the DDPM backbone (Ho et al., 2020) as the neural drift $f^\theta$ in our method, similar to D3PM. We report the

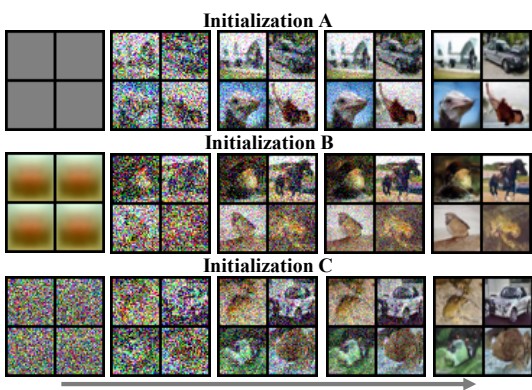

**Figure 4:** Integer-bridges can generate high-quality discrete samples with different initial distribution.

Inception Score (IS) Salimans et al. (2016), Fréchet Inception Distance (FID) Heusel et al. (2017) and negative log-likelihood (NLL) of the test dataset. We call our method *Integer-Bridge* in this case. The results are shown in Table 4 and Figure 4.

**Result** In Table 4, Integer-Bridge with Initialization A,B,C can all get lower FIDs ($\leq 7$) than the variants of D3PM. Among the three kinds of Integer-bridges, Integer-Bridge (Init. B) obtains the lowest NLL (3.35). It also beats D3PM Gauss + logistic (3.435) on NLL, which has the best NLL in the variants of D3PM.

## 5 CONCLUSION AND LIMITATIONS

We present a framework for learning diffusion generative models on constrained data domains. It leaves a number of directions for further explorations and improvement. For example, the practical impact of the choices of the bridges $\mathbb{Q}$, in terms of initialization, dynamics, and noise schedule, are still not well understood and need more systematical studies. Besides, our current method is limited to $\Omega$ that are factorizable and integrable. Moreover, application of $\Omega$-bridge to many other practical fields also needs investigation in the future.

## ACKNOWLEDGEMENTS

This research is supported by NSF CAREER1846421, SenSE2037267, EAGER-2041327, Office of Navy Research, and NSF AI Institute for Foundations of Machine Learning (IFML).

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

**Roadmap** The appendix is structured as follows:

- Appendix A provides the theoretical analysis and derivation of diffusion bridges. In particular, Appendix A.1 shows the derivation of the main training loss; Appendix A.2 derives the drift term $\eta^{\Pi^*}$ of $\mathbb{Q}^{\Pi^*}$; Appendix A.3 proves that we can actually use a Markov model $\mathbb{P}^\theta$ to match all time-marginals with $\mathbb{Q}^{\Pi^*}$; Appendix A.4 explains why we can use arbitrary initialization when constructing bridge processes and discusses the reciprocal structure of $\mathbb{Q}^{\Pi^*}$; Appendix A.5 provides analysis on the time-discretization error and statistical error of the practical discretized algorithm. Appendix A.6 and A.7 presents details on the condition and examples of $\Omega$-bridge construction.
- Appendix B shows additional experiment details and results.

## A  THEORETICAL ANALYSIS ON BRIDGES

### A.1  DERIVATION OF THE MAIN LOSS IN EQUATION (13)

[Proof of Equation (13)] Denote by $\mathbb{Q}^x = \mathbb{Q}(\cdot|Z_T = x)$. Note that

$$\mathcal{KL}(\mathbb{Q}^{\Pi^*} \,||\, \mathbb{P}^\theta) = \mathbb{E}_{x\sim\Pi^*, Z\sim\mathbb{Q}^x}\left[\log\frac{\mathrm{d}\mathbb{Q}^{\Pi^*}}{\mathrm{d}\mathbb{P}^\theta}(Z)\right]$$

$$= \mathbb{E}_{x\sim\Pi^*, Z\sim\mathbb{Q}^x}\left[\log\frac{\mathrm{d}\mathbb{Q}^x}{\mathrm{d}\mathbb{P}^\theta}(Z) + \log\frac{\mathrm{d}\mathbb{Q}^{\Pi^*}}{\mathrm{d}\mathbb{Q}^x}(Z)\right]$$

$$= \mathbb{E}_{x\sim\Pi^*}\left[\mathcal{KL}(\mathbb{Q}^x \,||\, \mathbb{P}^\theta)\right] + const,$$

where $const$ denotes a constant that is independent of $\theta$. Recall that $\mathbb{Q}^x$ follows $\mathrm{d}Z_t = \eta^x(Z_{[0,t]}, t)\mathrm{d}t + \sigma(Z_t, t)\mathrm{d}W_t$, and $\mathbb{P}^\theta$ follows $\mathrm{d}Z_t = s^\theta(Z_t, t)\mathrm{d}t + \sigma(Z_t, t)\mathrm{d}W_t$. By Girsanov theorem (e.g., Lejay, 2018),

$$\mathcal{KL}(\mathbb{Q}^x \,||\, \mathbb{P}^\theta) = \mathcal{KL}(\mathbb{Q}_0^x \,||\, \mathbb{P}_0^\theta) + \frac{1}{2}\mathbb{E}_{Z\sim\mathbb{Q}^x}\left[\int_0^T \left\|s^\theta(Z_t, t) - \eta^x(Z_{[0,t]}, t)\right\|_2^2 \mathrm{d}t\right]$$

$$= \mathbb{E}_{Z\sim\mathbb{Q}^x}\left[-\log p_0^\theta(Z_0) + \frac{1}{2}\int_0^T \left\|s^\theta(Z_t, t) - \eta^x(Z_{[0,t]}, t)\right\|_2^2 \mathrm{d}t\right] + const.$$

Hence

$$L(\theta) = \mathbb{E}_{x\sim\Pi^*, Z\sim\mathbb{Q}^x}\left[-\log p_0^\theta(Z_0) + \frac{1}{2}\int_0^T \left\|s^\theta(Z_t, t) - \eta^x(Z_{[0,t]}, t)\right\|_2^2 \mathrm{d}t\right] + const$$

$$= \mathbb{E}_{Z\sim\mathbb{Q}^{\Pi^*}}\left[-\log p_0^\theta(Z_0) + \frac{1}{2}\int_0^T \left\|s^\theta(Z_t, t) - \eta^{Z_T}(Z_{[0,t]}, t)\right\|_2^2 \mathrm{d}t\right] + const.$$

### A.2  DERIVATION OF THE DRIFT $\eta^{\Pi^*}$ OF $\mathbb{Q}^{\Pi^*}$

**Lemma A.1.** *Let $\mathbb{Q}^x$ is the law of*

$$\mathrm{d}Z_t^x = \eta^x(Z_{[0,t]}^x, t)\mathrm{d}t + \sigma(Z_t^x, t)\mathrm{d}W_t, \quad Z_0 \sim \mathbb{Q}_0^x,$$

*and $\mathbb{Q}^{\Pi^*} \coloneqq \int \mathbb{Q}^x(Z)\Pi^*(\mathrm{d}x)$ for a distribution $\Pi^*$ on $\mathbb{R}^d$. Then $\mathbb{Q}^{\Pi^*}$ is the law of*

$$\mathrm{d}Z_t = \eta^{\Pi^*}(Z_{[0,t]}, t)\mathrm{d}t + \sigma(Z_t, t)\mathrm{d}W_t, \quad Z_0 \sim \mathbb{Q}_0^{\Pi^*},$$

*where*

$$\eta^{\Pi^*}(z_{[0,t]}, t) = \mathbb{E}_{x\sim\Pi^*, Z\sim\mathbb{Q}^x}[\eta^x(Z_{[0,t]}, t) \mid Z_{[0,t]} = z_{[0,t]}], \quad \mathbb{Q}_0^{\Pi^*}(\mathrm{d}z_0) = \mathbb{E}_{x\sim\Pi^*}[\mathbb{Q}_0^x(\mathrm{d}z_0)].$$

[Proof] $\mathbb{Q}^{\Pi^*}$ is the solution of the following optimization problem:

$$\mathbb{Q}^{\Pi^*} = \arg\min_{\mathbb{P}}\left\{\mathcal{KL}(\mathbb{Q}^{\Pi^*} \,||\, \mathbb{P}) = \mathbb{E}_{x\sim\Pi^*}[\mathcal{KL}(\mathbb{Q}^x \,||\, \mathbb{P})] + const\right\}.$$

By Girsanov's Theorem (e.g., Lejay, 2018), any stochastic process $\mathbb{P}$ that has $\mathcal{KL}(\mathbb{Q}^x \,||\, \mathbb{P}) < +\infty$ (and hence is equivalent to $\mathbb{Q}^x$) has a form of $\mathrm{d}Z_t = \eta^{\Pi^*}(Z_{[0,t]}, t)\mathrm{d}t + \sigma(Z_t, t)\mathrm{d}W_t$ for some measurable function $\eta^{\Pi^*}$, and

$$\mathbb{E}_{x\sim\Pi^*}[\mathcal{KL}(\mathbb{Q}^x \,||\, \mathbb{P})]$$

$$= \mathbb{E}_{x\sim\Pi^*}[\mathcal{KL}(\mathbb{Q}_0^x \,||\, \mathbb{P}_0)] + \mathbb{E}_{x\sim\Pi^*,Z\sim\mathbb{Q}^x}\left[\frac{1}{2}\int_0^T \left\|\sigma(Z_t, t)^{-1}(\eta^{\Pi^*}(Z_{[0,t]}, t) - \eta^x(Z_{[0,t]}, 0))\right\|_2^2\right].$$

It is clear that to achieve the minimum, we need to take $\mathbb{P}_0(\cdot) = \mathbb{E}_{x\sim\Pi^*}[\mathbb{Q}_0^x(\cdot)]$ and $\eta^{\Pi^*}(z_{[0,t]}, t) = \mathbb{E}_{x\sim\Pi^*,Z\sim\mathbb{Q}^x}[\eta^x(Z_{[0,t]}, t) \mid Z_{[0,t]} = z_{[0,t]}]$, which yields the desirable form of $\mathbb{Q}^{\Pi^*}$.

### A.3   $\mathbb{P}^{\theta^*}$ Yields a Markovization of $\mathbb{Q}^{\Pi^*}$

As $\mathbb{P}^\theta$ is Markov by the model assumption, it can not perfectly fit $\mathbb{Q}^{\Pi^*}$ which is non-Markov in general. This is a substantial problem because $\mathbb{Q}^{\Pi^*}$ can be non-Markov *even if* $\mathbb{Q}^x$ is Markov for all $x \in \Omega$ (see Section A.4). In fact, using Doob's $h$-transform method (Doob, 1984), $\mathbb{Q}^{\Pi^*}$ can be shown to be the law of a diffusion process

$$\mathrm{d}Z_t = \eta^{\Pi^*}(Z_{[0,t]}, t)\mathrm{d}t + \sigma(Z_t, t)\mathrm{d}W_t, \quad \eta^{\Pi^*}(z_{[0,t]}, t) = \mathbb{E}_{Z\sim\mathbb{Q}^{\Pi^*}}\left[\eta^{Z_T}(z_{[0,t]}, t) \mid Z_{[0,t]} = z_{[0,t]}\right],$$

where $\eta^{\Pi^*}$ is the expectation of $\eta^x$ when $x = Z_T$ is drawn from $\mathbb{Q}$ conditioned on $Z_{[0,t]}$.

We resolve this by observing that it is not necessary to match the whole path measure ($\mathbb{P}^\theta \approx \mathbb{Q}^{\Pi^*}$) to match the terminal ($\mathbb{P}_T^\theta \approx \mathbb{Q}_T^{\Pi^*} = \Pi^*$). It is enough for $\mathbb{P}^\theta$ to be the best Markov approximation (a.k.a. Markovization) of $\mathbb{Q}^{\Pi^*}$, which matches all (hence terminal) fixed-time marginals with $\mathbb{Q}^{\Pi^*}$:

$$\mathrm{Proj}(\mathbb{Q}^{\Pi^*}, \mathcal{M}) \coloneqq \underset{\mathbb{P}\in\mathcal{M}}{\arg\min}\, \mathcal{KL}(\mathbb{Q}^{\Pi^*} \,||\, \mathbb{P}), \quad \mathcal{M} = \text{the set of all Markov processes on } [0, T].$$

**Proposition A.2.** *The global optimum of $\mathcal{L}(\theta)$ in (11) and (13) is achieved by $\theta^*$ if*

$$s^{\theta^*}(z, t) = \mathbb{E}_{Z\sim\mathbb{Q}^{\Pi^*}}\left[\eta^{Z_T}(Z_{[0,t]}, t) \mid Z_t = z\right], \quad \mu^{\theta^*}(\mathrm{d}z_0) = \mathbb{Q}_0^{\Pi^*} = \mathbb{E}_{x\sim\Pi^*}[\mathbb{Q}_0^x(\mathrm{d}z_0)]. \quad (14)$$

*In this case, $\mathbb{P}^{\theta^*} = \mathrm{Proj}(\mathbb{Q}^{\Pi^*}, \mathcal{M})$ is the Markovization of $\mathbb{Q}^{\Pi^*}$, with which it matches all time-marginals: $\mathbb{P}_t^{\theta^*} = \mathbb{Q}_t^{\Pi^*}$ for all time $t \in [0, T]$. In addition,*

$$\mathcal{KL}(\Pi^* \,||\, \mathbb{P}_T^\theta) \leq \mathcal{KL}(\mathbb{P}^{\theta^*} \,||\, \mathbb{P}^\theta) = \mathcal{KL}(\mathbb{Q}^{\Pi^*} \,||\, \mathbb{P}^\theta) - \mathcal{KL}(\mathbb{Q}^{\Pi^*} \,||\, \mathbb{P}^{\theta^*}) = \mathcal{L}(\theta) - \mathcal{L}(\theta^*). \quad (15)$$

Note that $s^{\theta^*}$ is a conditional expectation of $\eta^{\Pi^*}$: $s^{\theta^*}(z, t) = \mathbb{E}_{Z\sim\mathbb{Q}^{\Pi^*}}[\eta^{\Pi^*}(Z_{[0,t]}, t) \mid Z_t = z]$. Theorem 1 of Peluchetti (2021) gives a related result that the marginals of mixtures of Markov diffusion processes can be matched by another Markov diffusion process, but does not discuss the issue of Markovization nor connect to KL divergence. Theorem 1 of Song et al. (2021) is the special case of (15) when $\mathbb{Q}^{\Pi^*}$ is Markov.

[Proof of Proposition A.2] It is the combined result of Lemma A.3 and Lemma A.4 below.

**Lemma A.3.** *Let $\mathbb{Q}$ be a non-Markov diffusion process on $[0, T]$ of form*

$$\mathbb{Q}: \quad \mathrm{d}Z_t = \eta(Z_{[0,t]}, t)\mathrm{d}t + \sigma(Z_t, t)\mathrm{d}W_t, \quad Z_0 \sim \mathbb{Q}_0,$$

*and $\mathbb{M} = \arg\min_{\mathbb{P}\in\mathcal{M}} \mathcal{KL}(\mathbb{Q} \,||\, \mathbb{P})$ be the Markovization of $\mathbb{Q}$, where $\mathcal{M}$ is the set of all Markov processes on $[0, T]$. Then $\mathbb{Q}$ is the law of*

$$\mathbb{M}: \quad \mathrm{d}Z_t = m(Z_t, t)\mathrm{d}t + \sigma(Z_t, t)\mathrm{d}W_t, \quad Z_0 \sim \mathbb{Q}_0,$$

*where*

$$m(z, t) = \mathbb{E}_{Z\sim\mathbb{Q}}[\eta(Z_{[0,t]}, t) \mid Z_t = z].$$

*In addition, we have $\mathbb{Q}_t = \mathbb{M}_t$ for all time $t \in [0, T]$.*

[Proof] By Girsanov's Theorem (e.g., Lejay, 2018), any process that has $\mathcal{KL}(\mathbb{Q} \,||\, \mathbb{M}) < +\infty$ (and hence is equivalent to $\mathbb{Q}$) has a form of $\mathrm{d}Z_t = m(Z_{[0,t]}, t)\mathrm{d}t + \sigma(Z_t, t)\mathrm{d}W_t$, where $m$ is a measurable function. Since $\mathbb{M}$ is Markov, we have $m(Z_{[0,t]}, t) = m(Z_t, t)$. Then

$$\mathcal{KL}(\mathbb{Q} \,||\, \mathbb{P}) = \mathcal{KL}(\mathbb{Q}_0 \,||\, \mathbb{P}_0) + \mathbb{E}_{Z \sim \mathbb{Q}}\left[\frac{1}{2}\int_0^T \left\|\sigma(Z_t, t)^{-1}(\eta(Z_{[0,t]}, t) - m(Z_t, 0))\right\|_2^2\right].$$

It is clear that to achieve the minimum, we need to take $\mathbb{M}_0 = \mathbb{Q}_0$ and $m(z, t) = \mathbb{E}_{Z \sim \mathbb{Q}}[\eta(Z_{[0,t]}, t) \mid Z_t = z]$.

To prove $\mathbb{Q}_t = \mathbb{M}_t$, note that by the chain rule of KL divergence:

$$\mathcal{KL}(\mathbb{Q} \,||\, \mathbb{P}) = \mathcal{KL}(\mathbb{Q}_t \,||\, \mathbb{P}_t) + \mathbb{E}_{Z_t \sim \mathbb{Q}_t}[\mathcal{KL}(\mathbb{Q}(\cdot|Z_t) \,||\, \mathbb{P}(\cdot|Z_t))], \quad \forall t \in [0, T].$$

As the second term $\mathbb{P}(\cdot|Z_t)$ is independent of the choice of the marginal $\mathbb{P}_t$ at time $t \in [0, T]$, the optimum should be achieved by $\mathbb{M}$ only if $\mathbb{M}_t = \mathbb{Q}_t$.

**Lemma A.4.** *Let*

$$\begin{aligned}
\mathbb{Q}: \quad & \mathrm{d}Z_t = \eta(Z_{[0,t]}, t)\mathrm{d}t + \sigma(Z_t, t)\mathrm{d}W_t, \quad Z_0 \sim \mathbb{Q}_0 \\
\mathbb{M}: \quad & \mathrm{d}Z_t = m(Z_t, t)\mathrm{d}t + \sigma(Z_t, t)\mathrm{d}W_t, \quad Z_0 \sim \mathbb{Q}_0, \\
\mathbb{P}^\theta: \quad & \mathrm{d}Z_t = s^\theta(Z_t, t)\mathrm{d}t + \sigma(Z_t, t)\mathrm{d}W_t, \quad Z_0 \sim \mathbb{P}_0^\theta,
\end{aligned}$$

*where $\mathbb{M}$ is the Markovization of $\mathbb{Q}$ (see Lemma A.3). Then*

$$\mathcal{KL}(\mathbb{Q} \,||\, \mathbb{P}^\theta) = \mathcal{KL}(\mathbb{Q} \,||\, \mathbb{M}) + \mathcal{KL}(\mathbb{M} \,||\, \mathbb{P}^\theta).$$

*Hence, assume there exists $\theta^*$ such that $\mathbb{P}^{\theta^*} = \mathbb{M}$ and write $\mathcal{L}(\theta) := \mathcal{KL}(\mathbb{Q} \,||\, \mathbb{P}^\theta)$. We have*

$$\mathcal{KL}(\mathbb{Q}_T \,||\, \mathbb{P}_T^\theta) = \mathcal{KL}(\mathbb{M}_T \,||\, \mathbb{P}_T^\theta) \leq \mathcal{KL}(\mathbb{M} \,||\, \mathbb{P}^\theta) = \mathcal{L}(\theta) - \mathcal{L}(\theta^*).$$

[Proof] Note that

$\mathcal{KL}(\mathbb{M} \,||\, \mathbb{P}^\theta)$

$$= \mathcal{KL}(\mathbb{M}_0 \,||\, \mathbb{P}_0^\theta) + \frac{1}{2}\mathbb{E}_{Z_t \sim \mathbb{M}_t}\left[\int_0^T \left\|\sigma(Z_t, t)^{-1}(s^\theta(Z_t, t) - m(Z_t, t))\right\|_2^2\right]\mathrm{d}t$$

$$= \mathcal{KL}(\mathbb{M}_0 \,||\, \mathbb{P}_0^\theta) + \frac{1}{2}\int_0^T \mathbb{E}_{Z_t \sim \mathbb{M}_t}\left[\left\|\sigma(Z_t, t)^{-1}(s^\theta(Z_t, t) - m(Z_t, t))\right\|_2^2\right]\mathrm{d}t$$

$$= \mathcal{KL}(\mathbb{Q}_0 \,||\, \mathbb{P}_0^\theta) + \frac{1}{2}\int_0^T \mathbb{E}_{Z_t \sim \mathbb{Q}_t}\left[\left\|\sigma(Z_t, t)^{-1}(s^\theta(Z_t, t) - m(Z_t, t))\right\|_2^2\right]\mathrm{d}t \quad /\!/\mathbb{Q}_t = \mathbb{M}_t \; \forall t$$

$$= \mathcal{KL}(\mathbb{Q}_0 \,||\, \mathbb{P}_0^\theta) + \mathbb{E}_{Z \sim \mathbb{Q}}\left[\frac{1}{2}\int_0^T \left\|\sigma(Z_t, t)^{-1}(s^\theta(Z_t, t) - m(Z_t, t))\right\|_2^2 \mathrm{d}t\right]$$

$$= \mathcal{KL}(\mathbb{Q}_0 \,||\, \mathbb{P}_0^\theta) + \frac{1}{2}\left\|s^\theta - m\right\|_{\mathbb{Q},\sigma}^2,$$

where we define $\|f\|_{\mathbb{Q},\sigma}^2 = \mathbb{E}_{Z \sim \mathbb{Q}}\left[\frac{1}{2}\int_0^T \left\|\sigma(Z_t, t)^{-1}f(Z_t, t)\right\|_2^2 \mathrm{d}t\right].$

On the other hand,

$$\mathcal{KL}(\mathbb{Q} \,||\, \mathbb{P}^\theta) = \mathcal{KL}(\mathbb{Q}_0 \,||\, \mathbb{P}_0^\theta) + \mathbb{E}_{Z \sim \mathbb{Q}}\left[\frac{1}{2}\int_0^T \left\|\sigma(Z_t, t)^{-1}(s^\theta(Z_t, 0)) - \eta(Z_{[0,t]}, t)\right\|_2^2 \mathrm{d}t\right]$$

$$= \mathcal{KL}(\mathbb{Q}_0 \,||\, \mathbb{P}_0^\theta) + \frac{1}{2}\left\|s^\theta - \eta\right\|_{\mathbb{Q},\sigma}^2$$

$$\mathcal{KL}(\mathbb{Q} \,||\, \mathbb{M}) = \mathbb{E}_{Z \sim \mathbb{Q}}\left[\frac{1}{2}\int_0^T \left\|\sigma(Z_t, t)^{-1}(\eta(Z_{[0,t]}, t) - m(Z_t, 0))\right\|_2^2 \mathrm{d}t\right]$$

$$= \frac{1}{2}\left\|\eta - m\right\|_{\mathbb{Q},\sigma}^2.$$

Using Lemma A.5 with $a(z) = \sigma(z,t)^{-1} s^\theta(z,t)$, and $b(z_{[0,t]}) = \sigma(z,t)^{-1} \eta(z_{[0,t]},t)$, we have the following bias-variance decomposition:

$$\left\| \eta - s^\theta \right\|_{\mathbb{Q},\sigma}^2 = \left\| s^\theta - m \right\|_{\mathbb{Q},\sigma}^2 + \left\| \eta - m \right\|_{\mathbb{Q},\sigma}^2 .$$

Hence, $\mathcal{KL}(\mathbb{Q} \,||\, \mathbb{P}^\theta) = \mathcal{KL}(\mathbb{M} \,||\, \mathbb{P}^\theta) + \mathcal{KL}(\mathbb{Q} \,||\, \mathbb{M})$.

Finally, $\mathcal{KL}(M_T \,||\, \mathbb{P}_T^\theta) \leq \mathcal{KL}(\mathbb{M} \,||\, \mathbb{P}^\theta)$ is the direct result of the following factorization of KL divergence:

$$\mathcal{KL}(\mathbb{M} \,||\, \mathbb{P}^\theta) = \mathcal{KL}(\mathbb{M}_T \,||\, \mathbb{P}_T^\theta) + \mathbb{E}_{x \sim \mathbb{M}_T} \left[ \mathcal{KL}(\mathbb{M}_T(\cdot | Z_T = x) \,||\, \mathbb{P}_T^\theta(\cdot | Z_T = x)) \right].$$

**Lemma A.5.** *Let $(X,Y)$ be a random variable and $a(x)$, $b(x,y)$ are square integral functions. Let $m(x) = \mathbb{E}[b(X,Y) \mid X = x]$. We have*

$$\mathbb{E}[\|a(X) - b(X,Y)\|_2^2] = \mathbb{E}[\|a(X) - m(X)\|_2^2] + \mathbb{E}[\|b(X,Y) - m(X)\|_2^2].$$

[Proof]

$$\mathbb{E}[\|a(X) - b(X,Y)\|_2^2] = \mathbb{E}[\|a(X) - m(X) + m(X) - b(X,Y)\|_2^2]$$
$$= \mathbb{E}[\|a(X) - m(X)\|_2^2] + \mathbb{E}[\|m(X) - b(X,Y)\|_2^2] + 2\Delta,$$

where

$$\Delta = \mathbb{E}[(a(X) - m(X))^\top (m(X) - b(X,Y)))]$$
$$= \mathbb{E}[(a(X) - m(X))^\top \mathbb{E}[(m(X) - b(X,Y))|X]]$$
$$= \mathbb{E}[(a(X) - m(X))^\top (m(X) - m(X))] = 0.$$

## A.4 MARKOV AND RECIPROCAL PROPERTIES OF $\mathbb{Q}^{\Pi^*}$

**Mixture of Bridges and Initialization**  It is an immediate observation that the mixtures of a set of bridges are also bridges: let $\mathbb{Q}^{z,A}$ be a set of $A$-bridges indexed by a variable $z$, then $\mathbb{Q}^A := \int \mathbb{Q}^{z,A} \mu(\mathrm{d}z)$ is an $x$-bridge for any distribution $\mu$ on $z$.

A special case is to take the mixture of the conditional bridges in (5) starting from different deterministic initialization, which shows that we can obtain a valid $x$-bridge by equipping the same drift in (5) with essentially *any* initialization. Hence, the choices of the drift force and initialization in $\mathbb{Q}^x$ can be completely decouple.

**Proposition A.6.** *Let $\tilde{\mathbb{Q}}$ is a path measure and $\Omega_x$ is the set of $z$ for which $\tilde{\mathbb{Q}}^{z_0,x}(\cdot) := \tilde{\mathbb{Q}}(\cdot | Z_T = x, Z_0 = z_0)$ exists. Then $\mathbb{Q}^x := \int \tilde{\mathbb{Q}}^{z_0,x} \mu(\mathrm{d}z_0 \mid x)$ is an $x$-bridge, for any distribution $\mu$ on $\Omega \times \Omega$.*

[Proof of Proposition A.6] This is an obvious result. We have $\mathbb{Q}^{z_0,x}(Z_T = x) = 1$ by the definition of conditioned processes. Hence $\mathbb{Q}^x(Z_T = x) = \int \mathbb{Q}^{z_0,x}(Z_T = x) \mu(\mathrm{d}z_0 \mid x) = \int \mu(\mathrm{d}z_0 \mid x) = 1$.

**Markov and Reciprocal Properties of $\mathbb{Q}^{\Pi^*}$**  If $\mathbb{Q}^x$ is constructed as $\mathbb{Q}^x = \mathbb{Q}(\cdot | Z_T = x)$, it is easy to see that $\mathbb{Q}^{\Pi^*} := \int \mathbb{Q}^x(\cdot) \Pi^*(\mathrm{d}x)$ is Markov iff $\mathbb{Q}$ is Markov.

**Proposition A.7.** *Assume $\mathbb{Q}^x = \mathbb{Q}(\cdot \mid Z_T = x)$ and $\pi^*(z) := \frac{\mathrm{d}\Pi^*}{\mathrm{d}\mathbb{Q}_T}(z)$ exists and is positive everywhere. Then $\mathbb{Q}^{\Pi^*}$ is Markov, iff $\mathbb{Q}$ is Markov.*

[Proof of Proposition A.7] If $\mathbb{Q}^x = \mathbb{Q}(\cdot \mid Z_T = x)$, we have from the definition of $\mathbb{Q}^{\Pi^*}$:

$$\mathbb{Q}^{\Pi^*}(Z) = \mathbb{Q}(Z|Z_T)\Pi^*(Z_T) = \mathbb{Q}(Z)\pi^*(Z_T),$$

where $\pi^*(Z_T) = \frac{\mathrm{d}\Pi^*}{\mathrm{d}\mathbb{Q}_T}(Z_T)$. Therefore, $\mathbb{Q}^{\Pi^*}$ is obtained by multiplying a positive factor $\pi^*(Z_T)$ on the terminal state $Z_T$ of $\mathbb{Q}$. Hence $\mathbb{Q}^{\Pi^*}$ has the same Markov structure as that of $\mathbb{Q}$.

If $\mathbb{Q}^x$ is constructed from mixtures of bridges as above, the resulting $\mathbb{Q}^{\Pi^*}$ is more complex. In fact, simply varying the initialization $\mu$ in Proposition (A.6) can change the Markov structure of $\mathbb{Q}^{\Pi^*}$.

**Proposition A.8.** *Take $\mathbb{Q}^x$ to be the dynamics in* (7) *initialized from $Z_0 \sim \mathcal{N}(0, v_0)$. Assume $\sigma_t > 0$, $\forall t \in [0, T]$. Then $\mathbb{Q}^{\Pi^*}$ is Markov only when $v_0 = 0$, or $v_0 = +\infty$.*

[Proof of Proposition A.8] When taking $\mathbb{Q}^x$ to be the dynamics (7) initialized from $Z_0 \sim \mu_0 = \mathcal{N}(0, v_0)$, we have $\mathbb{Q}^x = \int \mu_0(\mathrm{d}z_0)\tilde{\mathbb{Q}}^{z_0, x}$, where $\tilde{\mathbb{Q}}^{z_0, x} = \tilde{\mathbb{Q}}(\cdot | Z_0 = z_0, Z_T = x)$ with $\tilde{\mathbb{Q}}$ following Brownian motion $\mathrm{d}Z_t = \mathrm{d}W_t$. Hence, we can write $\mathbb{Q}^{\Pi^*}(\mathrm{d}Z) = \tilde{\mathbb{Q}}(\mathrm{d}Z)r(Z_0, Z_T)$, where $r(z_0, z_T) = \frac{\mathrm{d}\mu_0 \otimes \Pi^*}{\mathrm{d}\tilde{\mathbb{Q}}_{0,T}}(z_0, z_T)$. From Léonard et al. (2014), $\mathbb{Q}^{\Pi^*}$ is Markov iff $r(x, z_0) = f(x)g(z_0)$ for some $f$ and $g$, which is not the case except the degenerated case ($v_0 = 0$ and $v_0 = +\infty$) because $\tilde{\mathbb{Q}}_{0,1}$ is not factorized.

On the other hand, when $v_0 = 0$, we have that $\mathbb{Q}^x = \mathbb{Q}(\cdot | Z_T = x)$ is the standard Brownian bridge and hence $\mathbb{Q}^{\Pi^*}$ is Markov following Proposition A.7. When $v_0 = +\infty$, as the case of SMLD, $\mathbb{Q}^{\Pi^*}$ is the law of $Z_t = \tilde{Z}_{T-t}$ with $\mathrm{d}\tilde{Z}_t = \mathrm{d}W_t$ and $\tilde{Z}_0 \sim \Pi^*$, which is also Markov.

The right characterization of $\mathbb{Q}^{\Pi^*}$ from Proposition (A.6) involves reciprocal processes (Léonard et al., 2014).

**Definition A.9.** *A process $Z$ with law $\mathbb{Q}$ on $[0, T]$ is said to be reciporcal if it can be written into $\mathbb{Q} = \int \tilde{\mathbb{Q}}^{z_0, z_T} \mu(\mathrm{d}z_0, \mathrm{d}z_T)$, where $\tilde{\mathbb{Q}}$ is a Markov process and $\tilde{\mathbb{Q}}^{z_0, z_T} = \tilde{\mathbb{Q}}(\cdot | Z_0 = z_0, Z_T = z_T)$, and $\mu$ is a probability measure on $\Omega \times \Omega$.*

**Proposition A.10.** *$\mathbb{Q}^{\Pi^*}$ is reciprocal iff $\mathbb{Q}^x = \int \tilde{\mathbb{Q}}^{z_0, x} \mu(\mathrm{d}z_0 \mid x)$ for a Markov $\tilde{\mathbb{Q}}$ and distribution $\mu$.*

[Proof of Proposition A.10] Note that

$$\mathbb{Q}^{\Pi^*}(\cdot) = \int \pi(\mathrm{d}x)\mathbb{Q}^x(\cdot) = \int \pi(\mathrm{d}x)\mu(\mathrm{d}z_0 \mid x)\tilde{\mathbb{Q}}^{z_0, x}(\cdot).$$

Hence if $\tilde{\mathbb{Q}}$ is Markov, $\mathbb{Q}^{\Pi^*}$ is reciprocal by Definition A.9.

On the other hand, if $\mathbb{Q}^{\Pi^*}$ is reciprocal, we have $\mathbb{Q}^{\Pi^*}(\cdot) = \int \mathbb{M}^{z_0, x}(\cdot)\mu(\mathrm{d}z_0, \mathrm{d}x)$ for some Markov process $\mathbb{M}$ and probability measure $\mu$ on $\Omega \times \Omega$. In this case, we have $\mathbb{Q}^x(\cdot) = \mathbb{Q}^{\Pi^*}(\cdot | Z_T = x) = \int \mathbb{M}^{z_0, x}(\cdot)\mu(\mathrm{d}z_0 \mid x)$, assuming it exits.

Intuitively, a reciprocal process can be viewed as connecting the head and tail of a Markov chain, yielding a single loop structure. A characteristic property is $\mathbb{Q}(X_{[s,t]} \in A \mid Z_{[0,s]}, Z_{[t,T]}) = \mathbb{Q}(X_{[s,t]} \in A \mid Z_s, Z_t)$, where $A$ is any event that occur between time $s$ and $t$. Solutions of the Schrodinger bridge problems are reciprocal processes (Léonard et al., 2014).

## A.5 PRACTICAL ALGORITHM AND ERROR ANALYSIS

In practice, we need to introduce empirical and numerical approximations in both training and inference phases. Denote by $\tau = \{\tau_i\}_{i=1}^{K+1}$ a grid of time points with $0 = \tau_1 < \tau_2 \ldots < \tau_{K+1} = T$. During training, we minimize an empirical and time-discretized surrogate of $\mathcal{L}(\theta)$ as follows

$$\hat{\mathcal{L}}(\theta) = \frac{1}{n} \sum_{i=1}^n \ell(\theta; Z^{(i)}, \tau^{(i)}), \qquad \ell(\theta; Z, \tau) := -\log p_0^\theta(Z_0) + \frac{1}{2K} \sum_{k=1}^K \Delta(\theta; Z, \tau_k), \qquad (16)$$

where $\Delta(\theta; Z, t) := \left\| \sigma^{-1}(Z_t, t)(s^\theta(Z_t, t) - \eta^x(Z_{[0,t]}, t)) \right\|^2$, and $\{Z^{(i)}\}$ is drawn from $\mathbb{Q}^{\Pi^*}$, and $\tau^{(i)}$ can be either a deterministic uniform grid of $[0, T]$, i.e., $\tau^{(i)} = \{i/K\}_{i=0}^K$, or drawn i.i.d. uniformly on $[0, T]$ (see e.g., Song et al. (2020b); Ho et al. (2020)). A subtle problem here is that the

variance of $\Delta(\theta; Z, t)$ grows to infinite as $t \uparrow T$. Hence, we should not include $\Delta(\theta; Z, T)$ at the end point $\tau^{K+1} = T$ into the sum in the loss $\ell(\theta, Z, \tau)$ to avoid variance exploding.

In the sampling phase, the continuous-time model $\mathbb{P}^\theta$ should be approximated numerically. A standard approach is the Euler-Maruyama method, which simulates the trajectory on a time grid $\tau$ by

$$\hat{Z}_{\tau_{k+1}} = \hat{Z}_{\tau_k} + \epsilon_k s^\theta(\hat{Z}_{\tau_k}, \tau_k) + \sqrt{\epsilon_k}\sigma(\hat{Z}_{\tau_k}, \tau_k)\xi_k, \quad \epsilon_k = \tau_{k+1} - \tau_k, \quad \xi_k \sim \mathcal{N}(0, I_d), \quad (17)$$

The final output is $\hat{Z}_T$. The following result shows the KL divergence between $\Pi^*$ and the distribution of $\hat{Z}_T$ can be bounded by the sum of the step size and the expected optimality gap $\mathbb{E}[\hat{\mathcal{L}}(\theta) - \hat{\mathcal{L}}(\theta^*)]$ of the time-discretized loss in (16).

### A.5.1  TIME-DISCRETIZATION ERROR ANALYSIS (PROPOSITION A.11)

**Proposition A.11.** *Assume $\Omega = \mathbb{R}^d$ and $\sigma(z, t) = \sigma(t)$ is state-independent. Take the uniform time grid $\tau^{\mathrm{unif}} := \{i\epsilon\}_{i=0}^K$ with step size $\epsilon = T/K$ in the sampling step (17). Assume $\sigma(t) > c > 0$, $\forall t$ and $\sigma(t)$ is piecewise constant w.r.t. time grid $\tau^{\mathrm{unif}}$. Let $\mathcal{L}_\epsilon(\theta) = \mathbb{E}_{Z \sim \mathbb{Q}^{\Pi^*}}[\ell(\theta; Z, \tau^{\mathrm{unif}})]$. Let $\mathbb{P}_T^{\theta,\epsilon}$ be the distribution of the resulting sample $\hat{Z}_T$. Let $\theta^*$ be an optimal parameter satisfying (14). Assume $C_0 := \sup_{z,t}\left(\left\|s^{\theta^*}(z,t)\right\|^2/(1+\|z\|^2), \mathrm{tr}(\sigma^2(z,t)), \mathbb{E}_{\mathbb{P}^{\theta^*}}[\|Z_0\|^2]\right) < +\infty$, and $\left\|s^{\theta^*}(z,t) - s^{\theta^*}(z',t')\right\|_2^2 \le L\left(\|z-z'\|^2 + |t-t'|\right)$ for $\forall z, z' \in \mathbb{R}^d$ and $t, t' \in [0, T]$. Then*

$$\sqrt{\mathcal{KL}(\Pi^* \,||\, \mathbb{P}_T^{\theta,\epsilon})} \le \sqrt{\mathcal{L}_\epsilon(\theta) - \mathcal{L}_\epsilon(\theta^*)} + O(\sqrt{\epsilon}).$$

We provide the analysis and proof for Proposition A.11 in the following text.

**Proposition A.12.** *Assume $\Omega = \mathbb{R}^d$ and $\sigma(z, t) = \sigma(t)$ is state-independent and $\sigma(t) > c > 0$, $\forall t \in [0, T]$. Take the uniform time grid $\tau^{\mathrm{unif}} := \{i\epsilon\}_{i=0}^K$ with step size $\epsilon = T/K$ in the sampling step (17). Let $\mathcal{L}_\epsilon(\theta) = \mathbb{E}_{Z \sim \mathbb{Q}^{\Pi^*}}[\ell_\epsilon(\theta; Z)]$ with*

$$\ell_\epsilon(\theta, Z_t) = -\log p_0^\theta(Z_0) + \frac{1}{2K}\sum_{k=1}^K \left\|\sigma_k^{-1}(s^\theta(Z_{t_k}, t_k) - \eta^{Z_T}(Z_{[0,t_k]}, t_k))\right\|_2^2,$$

*where $\epsilon > 0$ is a step size with $T = K\epsilon$ and $t_k = (k-1)\epsilon$, and $\sigma_k^2 := (t_{k+1} - t_k)^{-1}\int_{t_k}^{t_{k+1}}\sigma(t)^2\mathrm{d}t$. Let $\mathbb{P}_T^{\theta,\epsilon}$ be the distribution of the sample $\hat{Z}_T$ resulting from the following Euler method:*

$$\hat{Z}_{t_{k+1}} = \hat{Z}_{t_k} + \epsilon s^\theta(Z_{t_k}, t_k) + \sqrt{\epsilon}\sigma_k\xi_k,$$

*where $\xi_k \sim \mathcal{N}(0, I_d)$ is the standard Gaussian noise in $\mathbb{R}^d$. Let $\theta^*$ be an optimal parameter satisfying (14). Assume $C_0 := \sup_{z,t}\left(\left\|s^{\theta^*}(z,t)\right\|^2/(1+\|z\|^2), \mathrm{tr}(\sigma^2(z,t)), \mathbb{E}_{\mathbb{P}^{\theta^*}}[\|Z_0\|^2]\right) < +\infty$, and $s^{\theta^*}$ satisfies $\left\|s^{\theta^*}(z,t) - s^{\theta^*}(z',t')\right\|_2^2 \le L\left(\|z-z'\|^2 + |t-t'|\right)$ for $\forall z, z' \in \mathbb{R}^d$ and $t, t' \in [0, T]$. Then we have*

$$\sqrt{\mathcal{KL}(\Pi^* \,||\, \mathbb{P}_T^{\theta,\epsilon})} \le \sqrt{\mathcal{L}_\epsilon(\theta) - \mathcal{L}_\epsilon(\theta^*)} + O(\sqrt{\epsilon}).$$

[Proof of Proposition A.11] This is the result of Lemma A.13 below by noting that the $\hat{\mathbb{P}}^\theta$ there is equivalent to the Euler method above, and $\mathcal{L}_\epsilon(\theta) - \mathcal{L}_\epsilon(\theta^*) \le \tilde{\mathcal{L}}_\epsilon(\theta) - \tilde{\mathcal{L}}_\epsilon(\theta^*)$ (because $\sigma_k^{-2} = ((t_{k+1} - t_k)^{-1}\int_{t_k}^{t_{k+1}}\sigma(t)^2)^{-1} \le (t_{k+1} - t_k)^{-1}\int_{t_k}^{t_{k+1}}\sigma(t)^{-2}$).

**Lemma A.13.** *Let $h$ be a step size and $\epsilon = T/K$ for a positive integer $K$. For each $t \in [0, \infty)$, denote by $\lfloor t \rfloor_\epsilon = \max(\{k\epsilon: k \in \mathbb{N}\} \cap [0, t])$. Assume*

$$\begin{aligned}
\mathbb{Q}^{\Pi^*}: \quad &\mathrm{d}Z_t = \eta^{\Pi^*}(Z_{[0,t]}, t)\mathrm{d}t + \sigma(Z_t, t)\mathrm{d}W_t, \quad Z_0 \sim \mathbb{Q}_0 \\
\mathbb{P}^{\theta^*}: \quad &\mathrm{d}Z_t = s^{\theta^*}(Z_t, t)\mathrm{d}t + \sigma(Z_t, t)\mathrm{d}W_t, \quad Z_0 \sim \mathbb{Q}_0, \\
\mathbb{P}^\theta: \quad &\mathrm{d}Z_t = s^\theta(Z_t, t)\mathrm{d}t + \sigma(Z_t, t)\mathrm{d}W_t, \quad Z_0 \sim \mathbb{P}_0^\theta \\
\hat{\mathbb{P}}^\theta: \quad &\mathrm{d}Z_t = s^\theta(Z_{\lfloor t \rfloor_\epsilon}, t)\mathrm{d}t + \sigma(Z_t, t)\mathrm{d}W_t, \quad Z_0 \sim \mathbb{P}_0^\theta,
\end{aligned}$$

*where $\mathbb{P}^{\theta^*}$ is the Markovianization of $\mathbb{Q}^{\Pi^*}$, and $\hat{\mathbb{P}}^{\theta}$ is a discretized version of $\mathbb{P}^{\theta}$. Define*

$$\tilde{\mathcal{L}}_\epsilon(\theta) = \mathbb{E}_{\mathbb{Q}^{\Pi^*}} \left[ -\log p_0^\theta(Z_0) + \frac{1}{2} \int_0^T \left\| \sigma^{-1}(Z_t, t)(s^\theta(Z_{\lfloor t \rfloor_\epsilon}, \lfloor t \rfloor_\epsilon) - \eta^{Z_T}(Z_{[0, \lfloor t \rfloor_\epsilon]}, \lfloor t \rfloor_\epsilon)) \right\|^2 \mathrm{d}t \right].$$

*Assume the conditions of Lemma A.17 holds for $\mathbb{P}^{\theta^*}$, and $\sigma(z,t) \geq c > 0$ for all $z, t$, and $s^{\theta^*}$ satisfies $\left\| s^{\theta^*}(z,t) - s^{\theta^*}(z', t') \right\|_2^2 \leq L \left( \|z - z'\|^2 + |t - t'| \right)$ for $\forall z, z' \in \mathbb{R}^d$ and $t, t' \in [0, T]$. Then*

$$\sqrt{\mathcal{KL}(\mathbb{P}^{\theta^*} \| \hat{\mathbb{P}}^\theta)} \leq \sqrt{\tilde{\mathcal{L}}_\epsilon(\theta) - \tilde{\mathcal{L}}_\epsilon(\theta^*)} + C\sqrt{\epsilon},$$

*where $C$ is a constant that is independent of $\epsilon$.*

[Proof] Define $\|f\|_{\mathbb{Q}, \sigma}^2 = \mathbb{E}_{Z \sim \mathbb{Q}}[\int_0^T \|\sigma(Z, t) f(Z, t)\|^2]$ for convenient notation. Let $s_\epsilon^\theta(Z, t) = s^\theta(Z_{\lfloor t \rfloor_\epsilon}, \lfloor t \rfloor_\epsilon)$, and $\eta_\epsilon = \eta^{Z_T}(Z_{[0, \lfloor t \rfloor_\epsilon]}, \lfloor t \rfloor_\epsilon)$.

$$\mathcal{KL}(\mathbb{P}^{\theta^*} \| \hat{\mathbb{P}}^\theta)$$

$$= \mathcal{KL}(\mathbb{P}_0^{\theta^*} \| \hat{\mathbb{P}}_0^\theta) + \frac{1}{2} \left\| s^{\theta^*} - s_\epsilon^\theta \right\|^2$$

$$\leq \mathcal{KL}(\mathbb{P}_0^{\theta^*} \| \mathbb{P}_0^\theta) + \frac{1}{2} \left( (1 + \omega) \left\| s_\epsilon^\theta - s_\epsilon^{\theta^*} \right\|_{\mathbb{P}^{\theta^*}, \sigma}^2 + (1 + 1/\omega) \left\| s^{\theta^*} - s_\epsilon^{\theta^*} \right\|_{\mathbb{P}^{\theta^*}, \sigma}^2 \right)$$

$$:= (1 + \omega) I_1 + (1 + 1/\omega) I_2,$$

where $\omega > 0$ is any positive number and

$$I_1 := \frac{1}{1 + \omega} \mathcal{KL}(\mathbb{P}_0^{\theta^*} \| \mathbb{P}_0^\theta) + \frac{1}{2} \left\| s_\epsilon^\theta - s_\epsilon^{\theta^*} \right\|_{\mathbb{P}^{\theta^*}, \sigma}^2$$

$$= \frac{1}{1 + \omega} \mathcal{KL}(\mathbb{P}_0^{\theta^*} \| \mathbb{P}_0^\theta) + \frac{1}{2} \left\| s_\epsilon^\theta - s_\epsilon^{\theta^*} \right\|_{\mathbb{Q}^{\Pi^*}, \sigma}^2$$

$$\leq \mathcal{KL}(\mathbb{P}_0^{\theta^*} \| \mathbb{P}_0^\theta) + \frac{1}{2} \left\| s_\epsilon^\theta - s_\epsilon^{\theta^*} \right\|_{\mathbb{Q}^{\Pi^*}, \sigma}^2$$

$$\leq \mathcal{KL}(\mathbb{P}_0^{\theta^*} \| \mathbb{P}_0^\theta) + \frac{1}{2} \left( \left\| s_\epsilon^\theta - \eta_\epsilon^{Z_T} \right\|_{\mathbb{Q}^{\Pi^*}, \sigma}^2 - \left\| s_\epsilon^{\theta^*} - \eta_\epsilon^{Z_T} \right\|_{\mathbb{Q}^{\Pi^*}, \sigma}^2 \right) \qquad \text{//Lemma A.5}$$

$$= \tilde{\mathcal{L}}_\epsilon(\theta) - \tilde{\mathcal{L}}_\epsilon(\theta^*),$$

and

$$I_2 := \frac{1}{2} \left\| s^{\theta^*} - s_\epsilon^{\theta^*} \right\|_{\mathbb{P}^{\theta^*}, \sigma}^2$$

$$\leq \frac{L}{2} \mathbb{E}_{\mathbb{P}^{\theta^*}} \left[ \int_0^T \sigma^{-2}(Z_t, t) \left( \left\| Z_t - Z_{\lfloor t \rfloor_\epsilon} \right\|^2 + (t - \lfloor t \rfloor_\epsilon) \right) \mathrm{d}t \right]$$

$$\leq \frac{L}{2c^2} \mathbb{E}_{\mathbb{P}^{\theta^*}} \left[ \int_0^T ((Z_t - Z_{\lfloor t \rfloor_\epsilon})^2 + (t - \lfloor t \rfloor_\epsilon)) \mathrm{d}t \right]$$

$$\leq \frac{L}{2c^2} (C_{\mathbb{P}^{\theta^*}} + 1) \int_0^T (t - \lfloor t \rfloor_\epsilon) \mathrm{d}t \qquad \text{//Lemma A.17}$$

$$= \frac{L}{2c^2} (C_{\mathbb{P}^{\theta^*}} + 1) \frac{T\epsilon}{2}. \qquad \text{//Lemma A.15,}$$

where $C_{\mathbb{P}^{\theta^*}}$ is a constant depending on $\mathbb{P}^{\theta^*}$ that comes from Lemma A.17. Hence

$$\mathcal{KL}(\mathbb{P}^{\theta^*} \| \hat{\mathbb{P}}^\theta) \leq \inf_{\omega \geq 0} (1 + \omega) I_1 + (1 + 1/\omega) I_2$$

$$= (\sqrt{I_1} + \sqrt{I_2})^2$$

$$\leq \left( \sqrt{\tilde{\mathcal{L}}_\epsilon(\theta) - \tilde{\mathcal{L}}_\epsilon(\theta^*)} + \frac{1}{2} \sqrt{\frac{L}{c^2} (C_{\mathbb{P}^{\theta^*}} + 1) T\epsilon} \right).$$

This completes the proof.

**Lemma A.14.** *For any $a, b \in \mathbb{R}^d$, and $\omega \geq 0$,*

$$\|a + b\|_2^2 \leq (1 + \omega) \|a\|_2^2 + (1 + 1/\omega) \|b\|_2^2.$$

[Proof]

$$(1 + \omega) \|a\|_2^2 + (1 + 1/\omega) \|b\|_2^2 \geq \|a\|_2^2 + \|b\|_2^2 + 2a^\top b = \|a + b\|_2^2$$

**Lemma A.15.** *Assume $T \geq 0$, $\epsilon \geq 0$ and $T/\epsilon \in \mathbb{N}$. We have*

$$\int_0^T (t - \lfloor t \rfloor_\epsilon) \mathrm{d}t = \frac{T\epsilon}{2}.$$

[Proof]

$$
\begin{aligned}
\int_0^T (t - \lfloor t \rfloor_\epsilon) \mathrm{d}t &= \sum_{k=0}^{K-1} \int_0^\epsilon (hk + x - hk) \mathrm{d}x \\
&= \sum_{k=0}^{K-1} \int_0^\epsilon x \mathrm{d}x \\
&= K\epsilon^2/2 \\
&= T\epsilon/2.
\end{aligned}
$$

**Lemma A.16** (Grönwall's inequality). *Let $I$ denote an interval of the real line of the form $[a, \infty)$ or $[a, b]$ or $[a, b)$ with $a < b$. Let $\alpha, \beta$ and $u$ be real-valued functions defined on $I$. Assume that $\beta$ and $u$ are continuous and that the negative part of $\alpha$ is integrable on every closed and bounded subinterval of $I$.*

*(a) If $\beta$ is non-negative and if $u$ satisfies the integral inequality*

$$u(t) \leq \alpha(t) + \int_a^t \beta(s) u(s) \, \mathrm{d}s, \qquad \forall t \in I,$$

*which is true if*

$$u'(t) \leq \alpha'(t) + \beta(t) u(t).$$

*then*

$$u(t) \leq \alpha(t) + \int_a^t \alpha(s) \beta(s) \exp\left(\int_s^t \beta(r) \, \mathrm{d}r\right) \mathrm{d}s, \qquad t \in I.$$

*(b) If, in addition, the function $\alpha$ is non-decreasing, then*

$$u(t) \leq \alpha(t) \exp\left(\int_a^t \beta(s) \, \mathrm{d}s\right), \qquad t \in I.$$

**Lemma A.17.** *Consider*

$$\mathrm{d}X_t = b(Z_t, t) \mathrm{d}t + \sigma(Z_t, t) \mathrm{d}W_t, \qquad X_0 = 0, t \in [0, T].$$

*Assume there exists a finite constant $C_0$, such that*

$$\|b(x, t)\|_2^2 \leq C_0(1 + \|x\|_2^2), \quad \forall x \in \mathbb{R}^d, \ t \in [0, T],$$
$$\mathrm{tr}(\sigma\sigma^\top(x, t)) \leq C_0, \quad \forall x \in \mathbb{R}^d, t \in [0, T].$$

*and $\mathbb{E}[\|Z_0\|_2^2] \leq C_0$.*

*Then for any $0 \leq s \leq t \leq T$, we have*

$$\mathbb{E}[\|Z_t - Z_s\|_2^2] \leq K_{C_0, T}(t - s),$$

*where $K_{C_0, T}$ is a finite constant that depends on $C_0$ and $T$.*

[Proof] Let $\eta = \sup_{x,t} \operatorname{tr}(\sigma\sigma^\top(x,t))$. We have by Ito Lemma,

$$
\begin{aligned}
\frac{\mathrm{d}}{\mathrm{d}t}\mathbb{E}\left[\|Z_t - Z_s\|_2^2\right] &= \mathbb{E}\left[2(Z_t - Z_s)^\top(b(Z_t,t) + \mathrm{d}W_t) + \eta\right] \\
&= \mathbb{E}\left[2(Z_t - Z_s)^\top b(Z_t,t) + \eta\right] \\
&= \mathbb{E}\left[\|Z_t - Z_s\|_2^2 + \|b(Z_t,t)\|_2^2 + d\right] \\
&\leq \mathbb{E}\left[\|Z_t - Z_s\|_2^2 + C_0(1 + \|Z_t\|_2^2) + \eta\right] \\
&\leq (1 + 2C_0)\mathbb{E}\left[\|Z_t - Z_s\|_2^2\right] + \eta + C_0(1 + 2\mathbb{E}\left[\|Z_s\|_2^2\right]).
\end{aligned}
$$

Using Gronwall's inequality,

$$
\mathbb{E}\left[\|Z_t - Z_s\|_2^2\right] \leq (t-s)(\eta + C_0(1 + 2\mathbb{E}[\|Z_s\|_2^2]))\exp\left((t-s)(1 + 2C_0)\right).
$$

Taking $s = 0$ yields that

$$
\mathbb{E}\left[\|Z_t - Z_0\|_2^2\right] \leq t(\eta + C_0(1 + 2\mathbb{E}[\|Z_0\|_2^2]))\exp\left(t(1 + 2C_0)\right).
$$

Hence

$$
\begin{aligned}
\mathbb{E}[\|Z_t\|_2^2] &\leq 2\mathbb{E}\left[\|Z_t - Z_0\|_2^2\right] + 2\mathbb{E}\left[\|X_0\|_2^2\right] \\
&\leq 2t(\eta + C_0(1 + 2\mathbb{E}[\|Z_0\|_2^2]))\exp\left(t(1 + 2C_0)\right) + 2\mathbb{E}\left[\|Z_0\|_2^2\right] \\
&\leq 4T(C_0 + C_0^2)\exp(T(1 + 2C_0)) + 2C_0.
\end{aligned}
$$

Therefore,

$$
\begin{aligned}
&\mathbb{E}\left[\|Z_t - Z_s\|_2^2\right] \\
&\leq (t-s)(\eta + C_0(1 + 2\mathbb{E}[\|Z_s\|_2^2]))\exp\left((t-s)(1 + 2C_0)\right) \\
&\leq C(t-s),
\end{aligned}
$$

where

$$
C = (2C_0 + 4C_0^2 + 8C_0T(C_0 + C_0^2)\exp(T(1 + 2C_0)))\exp\left(T(1 + 2C_0)\right).
$$

### A.5.2 STATISTICAL ERROR ANALYSIS (PROPOSITION A.18)

To provide a simple analysis of the statistical error, we assume that $\hat{\theta}_n = \arg\min_\theta \hat{\mathcal{L}}(\theta)$ is an asymptotically normal M-estimator of $\theta^*$ following classical asymptotic statistics (Van der Vaart, 2000), with which we can estimate the rate of the excess risk $\mathcal{L}_\epsilon(\hat{\theta}_n) - \mathcal{L}_\epsilon(\theta^*)$ and hence the KL divergence.

**Proposition A.18.** *Assume the conditions in Proposition (A.11). Assume $\hat{\theta}_n = \arg\min_\theta \hat{\mathcal{L}}_\epsilon(\theta)$ with $\hat{\mathcal{L}}_\epsilon(\theta) = \sum_{i=1}^n \ell(\theta; Z^{(i)}, \tau^{\mathrm{unif}})/n$, $Z^{(i)} \sim \mathbb{Q}^{\Pi^*}$. Take $\mathbb{Q}^x$ to be the standard Brownian bridge $\mathrm{d}Z_t^x = \frac{x - Z_t^x}{T-t}\mathrm{d}t + \mathrm{d}W_t$ with $Z_0 \sim \mathcal{N}(0, v_0)$ and $v_0 > 0$. Assume $\sqrt{n}(\hat{\theta}_n - \theta^*) \xrightarrow{d} \mathcal{N}(0, \Sigma_*)$ as $n \to +\infty$, where $\Sigma_*$ is the asymptotic covariance matrix of the M estimator $\hat{\theta}_n$. Assume $\mathcal{L}_\epsilon(\theta)$ is second order continuously differentiable and strongly convex at $\theta^*$. Assume $\Pi^*$ has a finite covariance and admits a density function $\pi$ that satisfies $\sup_{t\in[0,T]} \mathbb{E}_{\mathbb{Q}^{\Pi^*}}\left[\left\|\nabla_\theta s^{\theta^*}(Z_t,t)\right\|^2 (1 + \|\nabla\log\pi(Z_T)\|^2 + \operatorname{tr}(\nabla^2\log\pi(Z_T)))\right] < +\infty$. We have*

$$
\mathbb{E}\left[\sqrt{\mathcal{KL}(\Pi^* \| \mathbb{P}_T^{\hat{\theta}_n,\epsilon})}\right] = O\left(\sqrt{\frac{\log(1/\epsilon) + 1}{n}} + \sqrt{\epsilon}\right). \tag{18}
$$

The expectation in Eq. (18) is w.r.t. the randomness of $\hat{\theta}_n$. The $\log(1/\epsilon)$ factor shows up as the sum of a harmonic series as the variance of $\Delta(\theta; Z, t)$ grows with $O(1/(T - t))$ when $t \uparrow T$. Taking $\epsilon = 1/n$ yields $\mathcal{KL}(\Pi^* \,||\, \mathbb{P}_T^{\hat{\theta}_n, \epsilon}) = O(\log n / n)$. If we want to achieve $\mathcal{KL}(\Pi^* \,||\, \mathbb{P}_T^{\hat{\theta}_n, \epsilon}) = O(\eta)$, it is sufficient to take $K = T/\epsilon = O(1/\eta)$ steps and $n = \O(\log(1/\eta)/\eta)$ data points.

[Proof of Proposition A.18] Let

$$\theta^* = \arg\min_\theta \mathcal{L}_\epsilon(\theta) := \mathbb{E}_{Z \sim \mathbb{Q}^{\Pi^*}}[\ell(\theta; Z)], \qquad \hat{\theta}_n = \arg\min_\theta \hat{\mathcal{L}}_\epsilon(\theta) := \frac{1}{n} \sum_{i=1}^n \ell(\theta; Z^{(i)}),$$

where $\{Z^{(i)}\}_{i=1}^n$ is drawn i.i.d. from $\mathbb{Q}^{\Pi^*}$. We assume that $\hat{\theta}_n$ is an asymptotically normal M-estimator, in which case we have

$$\sqrt{n}(\theta_n - \theta^*) \xrightarrow{d} \mathcal{N}(0, \Sigma_*),$$

where

$$\Sigma_* = H_*^{-1} V_* H_*^{-1}, \quad H_* = \mathbb{E}_{Z \sim \mathbb{Q}^{\Pi^*}} \left[ \nabla_{\theta\theta}^2 \ell(\theta^*; Z) \right], \quad V_* = \mathbb{E}[\nabla_\theta \ell(\theta^*; Z) \nabla_\theta \ell(\theta^*; Z)^\top],$$

and

$$n\mathbb{E}[(\mathcal{L}(\hat{\theta}_n) - \mathcal{L}(\theta^*))] \asymp \left[ \frac{1}{2} \sqrt{n}(\theta_* - \hat{\theta}_n)^\top H_* \sqrt{n}(\theta^* - \hat{\theta}_n) \right] \asymp \frac{1}{2} \mathrm{tr}(H_*^{-1} V_*),$$

where $f \asymp g$ denotes that $f - g = o(1)$. We now need to bound $\mathrm{tr}(H_*^{-1} V_*)$. Combining the results in Lemma A.19 and Lemma A.23, we have when $t_k = (k - 1)\epsilon$ and $T = K\epsilon$,

$$\mathrm{tr}(H_*^{-1} V_*) = O\left( 1 + \frac{1}{K} \sum_{k=1}^K \frac{1}{T - t_k} \right) = O(1 + \log(1/\epsilon)).$$

Hence,

$$\begin{aligned}
\mathbb{E}[\sqrt{\mathcal{KL}(\Pi^* \,||\, \mathbb{P}_T^{\hat{\theta}_n, \epsilon})}] &= O\left( \mathbb{E}[\sqrt{\mathcal{L}(\hat{\theta}_n) - \mathcal{L}(\theta^*)}] + \sqrt{\epsilon} \right) \\
&= O\left( \sqrt{\mathbb{E}[\mathcal{L}(\hat{\theta}_n) - \mathcal{L}(\theta^*)]} + \sqrt{\epsilon} \right) \\
&= O\left( \sqrt{\frac{\log(1/\epsilon + 1)}{n}} + \sqrt{\epsilon} \right).
\end{aligned}$$

**Lemma A.19.** *Assume the conditions in Proposition A.18. Define*

$$I_0 = \mathbb{E}_{Z \sim \mathbb{Q}^{\Pi^*}} \left[ \left\| \nabla \log p_0^{\theta^*}(Z_0) \right\|^2 \right], \quad I_k = \mathbb{E}_{Z \sim \mathbb{Q}^{\Pi^*}} \left[ \left\| \nabla_\theta s^{\theta^*}(Z_{t_k}, t_k) \right\|^2 \mathrm{tr}(\mathrm{cov}(\eta^{Z_T}(Z_{[0,t_k]}, t_k) \mid Z_{t_k})) \right],$$

*for $\forall k = 1, \dots K$. Then*

$$\mathrm{tr}(H_*^{-1} V_*)^{1/2} \leq \frac{1}{\lambda_{\min}(H_*)^{1/2}} \left( I_0^{1/2} + \left( \frac{1}{K} \sum_{k=1}^K I_k \right)^{1/2} \right).$$

[Proof] From Lemma A.21, $\mathrm{tr}(H_*^{-1} V_*) \leq (\lambda_{\min}(H_*))^{-1} \mathrm{tr}(V_*)$. Hence we just need to bound $\mathrm{tr}(V_*)$.

$$\operatorname{tr}(V_*)^{1/2} = \mathbb{E}_{Z \sim \mathbb{Q}^{\Pi^*}} \left[ \left\| \nabla_\theta \ell(\theta^*, Z) \right\|_2^2 \right]^{1/2}$$

$$\leq \mathbb{E}_{Z \sim \mathbb{Q}^{\Pi^*}} \left[ \left\| \nabla_\theta \ell(\theta^*, Z) \right\|_2^2 \right]^{1/2} + \frac{1}{K} \sum_{k=1}^K \mathbb{E}_{Z \sim \mathbb{Q}^{\Pi^*}} \left[ \left\| \nabla_\theta s^{\theta^*}(Z_{t_k}, t_k)(s^{\theta^*}(Z_{t_k}, t_k) - \eta^{Z_T}(Z_{[0,t_k]}, t_k)) \right\|_2^2 \right]^{1/2}$$

$$\leq \mathbb{E}_{Z \sim \mathbb{Q}^{\Pi^*}} \left[ \left\| \nabla_\theta \ell(\theta^*, Z) \right\|_2^2 \right]^{1/2} + \frac{1}{K} \sum_{k=1}^K \mathbb{E}_{Z \sim \mathbb{Q}^{\Pi^*}} \left[ \left\| \nabla_\theta s^{\theta^*}(Z_{t_k}, t_k) \right\|_2^2 \left\| (s^{\theta^*}(Z_{t_k}, t_k) - \eta^{Z_T}(Z_{[0,t_k]}, t_k)) \right\|_2^2 \right]^{1/2}$$

$$= \mathbb{E}_{Z \sim \mathbb{Q}^{\Pi^*}} \left[ \left\| \nabla_\theta \ell(\theta^*, Z) \right\|_2^2 \right]^{1/2} + \frac{1}{K} \sum_{k=1}^K \mathbb{E}_{Z \sim \mathbb{Q}^{\Pi^*}} \left[ \left\| \nabla_\theta s^{\theta^*}(Z_{t_k}, t_k) \right\|_2^2 \operatorname{tr}\left( \operatorname{cov}\left( \eta^{Z_T}(Z_{[0,t_k]}, t_k) \mid Z_{t_k} \right) \right) \right]^{1/2}$$

$$= I_0^{1/2} + \frac{1}{K} \sum_{k=1}^K I_k^{1/2}$$

$$\leq I_0^{1/2} + \sqrt{\frac{1}{K} \sum_{k=1}^K I_k}.$$

**Lemma A.20.** *Assume the results in Lemma A.19 and Lemma A.23 hold. Assume*

$$\max_{k \in 1, \ldots, K} \mathbb{E}_{Z \sim \Pi^*} \left[ \left\| \nabla_\theta \tilde{s}^{\theta^*}(Z_{t_k}, t_k) \right\|_2^2 \left( 1 + \|\nabla \log \pi^*(Z_T)\|_2^2 + \operatorname{tr}(\nabla^2 \log \pi^*(Z_T)) \right) \right] < +\infty,$$

*Then for $k = 1, \ldots, K$, we have $I_k = O\left(\frac{1}{T - t_k} + 1\right)$.*

[Proof] It is a direction application of (20).

**Lemma A.21.** *Let $A$ and $B$ be two $d \times d$ positive semi-definite matrices. Then $\operatorname{tr}(AB) \leq \lambda_{\max}(A)\operatorname{tr}(B)$.*

[Proof] Write $A$ into $A = \sum_{i=1}^d \lambda_i u_i u_i^\top$ where $\lambda_i$ and $u_i$ is the $i$-th eigenvalue and eigenvectors of $A$, respectively. Then

$$\operatorname{tr}(AB) = \operatorname{tr}(\sum_{i=1}^d \lambda_i u_i^\top B u_i) \leq \lambda_{\max}(A)\operatorname{tr}(\sum_{i=1}^d u_i^\top B u_i) = \lambda_{\max}(A)\operatorname{tr}(B).$$

**Controlling the Conditional Variance of the Regression Problem** Assume $\mathbb{Q}^x$ is the standard Brownian bridge:

$$\mathbb{Q}^x: \quad dZ_t^x = \frac{x - Z_t^x}{T - t} dt + dW_t, \quad Z_0 \sim \mathcal{N}(0, v_0). \tag{19}$$

In this case, the (ideal) loss function is

$$\mathcal{L}(\theta) = -\mathbb{E}_{X \sim \Pi^*, Z \sim \mathbb{Q}^X} \left[ \log p_0^\theta(Z_0) + \frac{1}{2} \int_0^T \left\| s^\theta(Z_t, t) - Y_t \right\|^2 dt \right], \quad \text{where } Y_t = \frac{X - Z_t}{1 - t}.$$

The second part of the loss is a least square regression for predicting $Y_t = \eta^X(Z_t, t)$ with $s^\theta(Z_t, t)$. The conditioned variance $\operatorname{cov}(Y_t \mid Z_t)$ is an important factor that influences the error of the regression problem. We now show that $\operatorname{tr}(\operatorname{cov}(Y_t \mid Z_t)) = O(1/T - t)$ which means that it explodes to infinity when $t \uparrow T$.

First, note that $\operatorname{tr}(\operatorname{cov}(Y_t \mid Z_t)) = \frac{1}{(T-t)^2} \operatorname{tr}(\operatorname{cov}(X \mid Z_t))$. Using the estimate in Lemma A.23, we have

$$\operatorname{tr}(\operatorname{cov}(Y_t \mid Z_t)) = O\left( \frac{1}{T - t} + \mathbb{E} \left[ \|\nabla_x \log \pi^*(X)\|_2^2 + \operatorname{tr}\left( \nabla^2 \log \pi^*(X) \right) \Big| Z_t \right] \right). \tag{20}$$

**Lemma A.22.** *For the standard Brownian bridge in* (19)*, we have*

$$Z_t^x \sim \mathcal{N}\left(\frac{t}{T}x, \ \frac{t(T-t)}{T} + \frac{(T-t)^2}{T^2}v_0\right).$$

[Proof] Let $Z_t^{z_0,x}$ be the same process that is initialized from $Z_0^{z_0,x} = z_0$. We have from the textbook result regarding Brownian bridge that we can write $Z_t^{z_0,x} = \frac{tx + (T-t)z_0}{T} + \sqrt{\frac{t(T-t)}{T}}\xi_t$ where $\xi_t$ is some standard Gaussian random variable. The result follows directly as $Z_t^x = Z_t^{Z_0,x}$ with $Z_0 \sim \mathcal{N}(0, v_0)$.

**Lemma A.23.** *Let $\pi^*$ be the density function $\Pi^*$ on $\mathbb{R}^d$ whose covariance matrix exists. When $X \sim \Pi^*$ and $Z \sim \mathbb{Q}^X$ from* (19) *with $v_0 > 0$. Then the density function $\rho_t(x|z)$ of $X|Z_t = z_t$ satisfies*

$$\rho_t(x|z) \propto \pi^*(x)\exp\left(-\frac{\left\|\frac{T}{t}z - x\right\|_2^2}{2\left(\frac{T(T-t)}{t} + v_0\frac{(T-t)^2}{t^2}\right)}\right). \tag{21}$$

*In addition, there exists positive constants $c < +\infty$ and $\tau \in (0, T)$, such that*

$$\mathrm{tr}(\mathrm{cov}_{\rho_t}(x|z)) \leq \begin{cases} w_t d + w_t^2 \mathbb{E}_{\rho_t}\left[\|\nabla_x \log \pi^*(x)\|_2^2 + \mathrm{tr}\left(\nabla^2 \log \pi^*(x)\right) \ \Big| \ z\right], & \text{when } \tau \leq t \leq T \\ c, & \text{when } 0 \leq t \geq \tau, \end{cases}$$

*where $w_t = \frac{T(T-t)}{t} + v_0\frac{(T-t)^2}{t^2}$. So $\mathrm{tr}(\mathrm{cov}_{\rho_t}(x|z))$ is bounded and decay to zero with rate $O(T-t)$ as $t \uparrow T$.*

[Proof] We know that $X \sim \Pi^*$ and $Z_t^X|X \sim \mathcal{N}(t/TX, \ w_t)$. Hence, (21) is a direct result of Bayes rule. Then Lemma A.25 gives

$$\mathrm{tr}(\mathrm{cov}_{\rho_t}(x|z)) = w_t d + w_t^2 \mathbb{E}_{\rho_t}\left[\|\nabla_x \log \pi^*(x)\|_2^2 + \mathrm{tr}\left(\nabla^2 \log \pi^*(x)\right) \Big| z\right].$$

On the other hand,

$$\rho_t(x|z) \propto \pi^*(x)\exp(-\frac{1}{2w_t}\|x\|^2 + \frac{T}{tw_t}z^\top x),$$

When $t \to 0$, we have $1/w_t \to 0$ and $T/(tw_t) \to 0$. Hence, $\rho_t(x|z)$ converges to $\pi^*(x)$ as $t \to 0$, as a result, $\mathrm{tr}(\mathrm{cov}_{\rho_t}(x|z)) \to \mathrm{tr}(\mathrm{cov}_{\pi^*}(x)) < +\infty$. Therefore, for any $c > 0$, there exists $t_0 > 0$, such that $\mathrm{tr}(\mathrm{cov}_{\rho_t}(x|z)) \leq \mathrm{tr}(\mathrm{cov}_{\pi^*}(x)) + c$ when $0 \leq t \leq t_0$.

**Remark A.24.** *We need to have $v_0 > 0$ to ensure that $T/(tw_t) \to 0$ in the proof of Lemma A.23. This is purely a technical reason, for yielding a finite bound of the conditioned variance when $t$ is close to $0$. We can establish the same result when $v_0 = 0$ by adding the assumption that $\max_{k \in 1,\ldots,K} \mathbb{E}_{Z \sim \mathbb{Q}^{\Pi^*}}\left[\left\|\nabla_\theta s^{\theta^*}(Z_{t_k}, t_k)\right\|_2^2 \mathrm{tr}(\mathrm{cov}_{\Pi^*_{Z_{t_k}}}(Z_T))\right] < +\infty$, where $\Pi^*_z$ is the distribution with density $\pi^*_z(x) \propto \pi^*(x)\exp(z^\top x/T)$.*

**Lemma A.25.** *Let $p(x) \propto \pi(x)\exp\left(-\alpha\frac{\|x-b\|_2^2}{2}\right)$ be a positive probability density function on $\mathbb{R}^d$, where $\alpha > 0$, $b \in \mathbb{R}$ and $\log \pi$ is continuously second order differentiable. Then*

$$\mathrm{tr}(\mathrm{cov}_p(x)) \leq \mathbb{E}_p[\|x\|_2^2] = \alpha^{-1}d + \alpha^{-2}\left(\mathbb{E}_p[\|\nabla_x \log \pi(x)\|_2^2 + \mathrm{tr}(\nabla^2 \log \pi(x))]\right).$$

[Proof] Let us focus on the case when $b = 0$ first. Stein's identity says that

$$\mathbb{E}_p\left[(\nabla_x \log \pi(x) - \alpha x)^\top \phi(x) + \nabla_x^\top \phi(x)\right] = 0,$$

for a general continuously differentiable function $\phi$ when the integrals above are finite.

Taking $\phi = x$ yields that

$$\mathbb{E}_p \left[ (\nabla_x \log \pi(x) - \alpha x)^\top x + d \right] = 0,$$

which gives

$$\mathbb{E}_p[\|x\|_2^2] = \alpha^{-1}(\mathbb{E}_p[\nabla_x \log \pi(x)^\top x] + d).$$

On the other hand, taking $\phi(x) = \nabla_x \log \pi(x)$ yields

$$\mathbb{E}_p \left[ (\nabla_x \log \pi(x) - \alpha x)^\top \nabla_x \log \pi(x) + \text{tr}(\nabla^2 \log \pi(x)) \right] = 0,$$

which gives

$$\mathbb{E}_p[\nabla_x \log \pi(x)^\top x] = \alpha^{-1} \left( \mathbb{E}_p[\|\nabla_x \log \pi(x)\|_2^2 + \text{tr}(\nabla^2 \log \pi(x))] \right).$$

This gives

$$\mathbb{E}_p[\|x\|_2^2] = d\alpha^{-1} + \alpha^{-2} \left( \mathbb{E}_p[\|\nabla_x \log \pi(x)\|_2^2 + \text{tr}(\nabla^2 \log \pi(x))] \right).$$

For $b \neq 0$, define $\tilde{p}(x) \propto \pi(x + b) \exp\left( -\frac{\alpha}{2} \|x\|^2 \right)$, which is the distribution of $\tilde{x} = x - b$ when $x \sim p$. Then applying the result above to $\tilde{p}$ yields

$$\begin{aligned}
\text{tr}(\text{cov}_p(x)) &= \text{tr}(\text{cov}_{\tilde{p}}(x)) \\
&\leq \alpha^{-1}d + \alpha^{-2}\mathbb{E}_{x \sim \tilde{p}} \left[ \|\nabla_x \log \pi(x + b)\|_2^2 + \text{tr}\left( \nabla^2 \log \pi(x + b) \right) \right] \\
&= \alpha^{-1}d + \alpha^{-2}\mathbb{E}_{\sim p} \left[ \|\nabla_x \log \pi(x)\|_2^2 + \text{tr}\left( \nabla^2 \log \pi(x) \right) \right].
\end{aligned}$$

## A.6 CONDITION FOR $\Omega$-BRIDGES

We provide the proof for Proposition 2.3.

[Proof of Proposition 2.3] By the formula of KL divergence between two diffusion processes, we have

$$\mathcal{KL}(\mathbb{Q}^\Omega \,\|\, \mathbb{P}^\theta) = \mathcal{KL}(\mathbb{Q}_0^\Omega \,\|\, \mathbb{P}_0^\theta) + \frac{1}{2}\mathbb{E}_{Z \sim \mathbb{Q}^\Omega} \left[ \int_0^T \|f^\theta(Z_t, t)\|_2^2 \, dt \right] < +\infty.$$

This means that $\mathbb{Q}^\Omega$ and $\mathbb{P}^\theta$ are absolutely continuous to each other, and hence have the same support. Therefore, $\mathbb{Q}^\Omega(Z_T \in \Omega) = 1$ implies that $\mathbb{P}^\theta(Z_T \in \Omega) = 1$.

## A.7 EXAMPLES OF $\Omega$-BRIDGES

If $\Omega$ is a product space, the integration can be factorized into one-dimensional integrals. Specifically, assume $\Omega = I_1 \times \cdots I_d$, then

$$\eta^\Omega(z, t) = \left[ \eta^{I_i}(z_i, t) \right]_{i=1}^d,$$

where $\eta^{I_i}$ is the drift fore of the $I_i$-bridge, and $z_i$ is the $i$-th element of $z = [z_i]$. Therefore, it is sufficient to focus on 1D case below.

Consider the bridge process constructed from the Brownian motion in (9). If $\Omega$ is a discrete set, say $\Omega = \{e_1 \ldots, e_K\}$, we have

$$\begin{aligned}
\eta^\Omega(z, t) &= \sigma_t^2 \frac{1}{\sum_{k=1}^K \omega(e_k, z, t)} \sum_{k=1}^K \omega(e_k, z, t) \frac{e_k - z}{\beta_T - \beta_t} \\
&= \sigma_t^2 \nabla_z \log \sum_{k=1}^K \omega(e_k, z, t),
\end{aligned}$$

| age | workclass | fnlwgt | education | marital-status | occupation | relationship | race | sex | capital-gain | capital-loss | hours-per-week | native-country | income |
|-----|-----------|--------|-----------|----------------|------------|--------------|------|-----|--------------|--------------|----------------|----------------|--------|
| 55 | Self-emp-not-inc | 229791 | Bachelors | Divorced | Prof-specialty | Other-relative | White | Male | 2561 | 77 | 37 | Cuba | >50K |
| 45 | Self-emp-not-inc | 313479 | Some-college | Never-married | Machine-op-inspct | Husband | White | Male | 4660 | 483 | 38 | United-States | >50K |
| 81 | Without-pay | 106624 | 9th | Married-civ-spouse | Adm-clerical | Husband | Black | Male | 1910 | 66 | 24 | United-States | <=50K |
| 41 | Private | 285037 | 9th | Divorced | Adm-clerical | Husband | White | Male | 3754 | 46 | 28 | Poland | <=50K |
| 22 | Private | 21419 | Bachelors | Widowed | Exec-managerial | Own-child | White | Female | 4689 | 209 | 34 | United-States | <=50K |

**Figure 5:** Generated tabular data from Mixed-Bridge.

where

$$\omega(e_k, z, t) = \exp\left(-\frac{\|z - e_k\|^2}{2(\beta_T - \beta_t)}\right).$$

If $\Omega = [a, b]$, we have

$$\eta^\Omega(z, t) = \sigma_t^2 \frac{1}{\int_a^b \omega(e, z, t)} \int_a^b \omega(e, z, t) \frac{e - z}{\beta_T - \beta_t} \mathrm{d}e$$

$$= \sigma_t^2 \nabla_z \log \int_a^b \omega(e, z, t) \mathrm{d}e$$

$$= \sigma_t^2 \nabla_z \log\left(F(\frac{z - a}{\sqrt{\beta_T - \beta_t}}) - F(\frac{z - b}{\sqrt{\beta_T - \beta_t}})\right),$$

where $F$ is the standard Gaussian CDF.

## B  ADDITIONAL MATERIALS OF THE EXPERIMENTS

In our experiments, $T = 1$ and $\epsilon = T/K = 1/K$. Moreover, we take the time grid by randomly sampling from $\{i/K\}_{i=0}^{K-1}$ for the training objective Eq. (13). For evaluation, we calculate the standard evidence lower bound (ELBO) by viewing the resulting time-discretized model as a latent variable model:

$$\mathbb{E}_{X \sim \Pi^*}[-\log \hat{p}_T^\theta(X)] \leq \mathbb{E}_{Z \sim \mathbb{Q}^{\Pi^*}}\left[-\log \frac{\hat{p}_0^\theta(Z_0)}{q_0(Z_0)} - \sum_{k=1}^K \log \frac{\hat{p}_{t_{k+1}|t_k}^\theta(Z_{t_{k+1}}|Z_{t_k})}{q_{t_{k+1}|t_k}(Z_{t_{k+1}}|Z_{t_k})}\right],$$

where $t_k = (k-1)\epsilon$, and $\hat{p}^\theta$ is the density function of the time-discretized version of $\mathbb{P}^\theta$, and $q$ is the density function of $\mathbb{Q}$. We adopt Monte-Carlo sampling to estimate the log-likelihood. As in (Song et al., 2020b), we repeat 5 times in the test set for the estimation. For categorical/integer/grid generation, the likelihood of the last step should take the rounding into account: in practice, we have $\hat{Z}_T = \mathrm{rounding}(\hat{Z}_{t_K} + \epsilon s^\theta(\hat{Z}_{t_K}, t_K) + \sqrt{\epsilon}\sigma(Z_{t_k}, t_K)\xi_K, \Omega)$, where $\mathrm{rounding}(x, \Omega)$ denotes finding the nearest element of $x$ on $\Omega$, and hence the likelihood $\hat{p}_{T|t_K}^\theta$ of the last step should incorporate the rounding operator as a part of the model.

### B.1  GENERATING MIXED-TYPE TABULAR DATA

In this experiment, the metrics are measured by the implementation from Synthetic Data Vault (SDV) (Patki et al., 2016). For baseline methods, we adopt their open-sourced official implementation [1]. For the machine learning models adopted for evaluation, logistic regerssion, AdaBoost and MLP, we directly use their default configuration in SDV. For the results in Table 1, we repeat the experiments with 5 different random seeds and report their standard deviation. We provide additional generated samples from Mixed-Bridge in Figure 5.

---

[1] https://github.com/sdv-dev/CTGAN

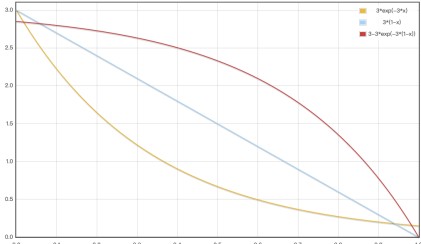

**Figure 6:** Visualization of the noise schedule of Noise decay A, Noise decay B and Noise decay C.

## B.2 GENERATING INTEGER-VALUED POINT CLOUDS

In this experiment, we need to process point cloud data on integer grid. To prepare the data, we firstly sample 2048 points from the ground truth mesh. Then, we normalize all the point clouds to a unit bounding box. After this, we simply project the points onto grid point by rounding the coordinate to integer. The metrics in the main text, MMD, COV and 1-NNA are computed with respect to the post-processed integer-valued training point clouds. For the results in Table 2, we repeat the experiments for 3 times and report the mean of the experiments.

## B.3 GENERATING SEMANTIC SEGMENTATION MAPS ON CITYSCAPES

In this experiment, we set *(Noise Decay A)*: $\sigma_t^2 = 3\exp(-3t)$; *(Noise Decay B)*: $\sigma_t^2 = 3(1-t)$; *(Noise Decay C)* $\sigma_t^2 = 3 - 3\exp(-3(1-t))$. We visualize the noise schedule in Figure 6. Note that, except for Constant Noise, all the other three processes gradually decrease the magnitude of the noise as $t \to 1$. For fair comparison, we use the same neural network as in Hoogeboom et al. (2021). The network is optimized with Adam optimizer with a learning rate of $0.0002$. The model is trained for 500 epochs. The CityScapes dataset (Cordts et al., 2016) contains photos captured by the cameras on the driving cars. A pixel-wise semantic segmentation map is labeled for each photo. As in (Hoogeboom et al., 2021), we rescale the segmentation maps from cityscapes to $32 \times 64$ images using nearest neighbour interpolation. Our training set and test set is exactly the same as that of (Hoogeboom et al., 2021) for fair comparison. For the results in Table 3,we repeat the experiments for 3 times and report the mean of the experiments. We provide more samples in Figure 9.

## B.4 DISCRETE CIFAR10 GENERATION

The model is trained using the same training strategy as DDPM (Ho et al., 2020) with the code base provided in Song et al. (2020b). Specifically, the neural network is the same U-Net structure as the implementation in Song et al. (2020b). The optimizer is Adam with a learning rate of $0.0002$. According to common practice (Song & Ermon, 2020; Song et al., 2020b), the training is smoothed by exponential moving average (EMA) with a factor of $0.999$. We use $K = 1000$ and $\mathrm{d}t = 0.001$ for discretizing the SDE. To account for the discretization error, after the final step, we apply rounding to the generated images to get real integer-valued images. We compare the value distribution of the generated images in Figure 8.

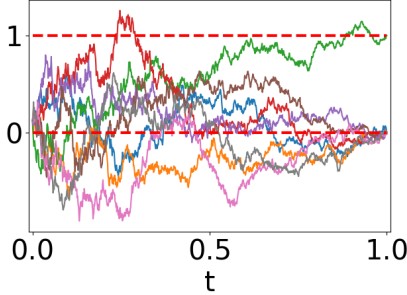

**Figure 7:** Diffusion process of one pixel (a 8-dimensional vector) in CityScapes. As $t \to 1$, 7 of the dimensions reaches 0, while 1 of the dimensions reaches 1, turning the vector into a `one-hot` vector.

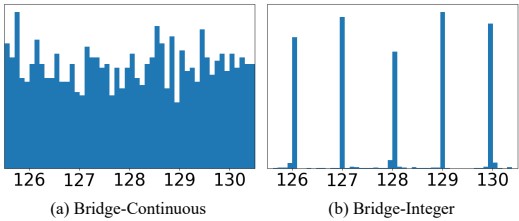

(a) Bridge-Continuous      (b) Bridge-Integer

**Figure 8:** Final value distribution of the generated images with Bridge-Continuous and Bridge-Integer (before rounding) on CIFAR10. We only show the values in $[125.5, 130.5]$ for visual clarity. Integer-Bridge generates discrete values.

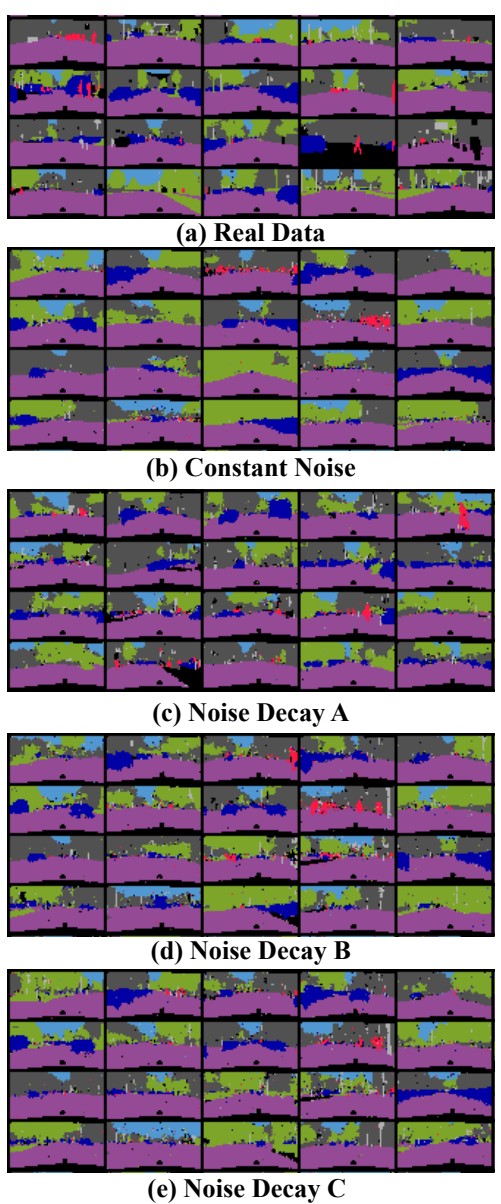

(a) Real Data

(b) Constant Noise

(c) Noise Decay A

(d) Noise Decay B

(e) Noise Decay C

**Figure 9:** Additional samples from real data, Constant Noise, Noise decay A, Noise decay B and Noise decay C.

