# OpenReview forum: "Learning Diffusion Bridges on Constrained Domains"
_ICLR.cc/2023/Conference — ICLR 2023 notable top 25%_

### Official Review · Reviewer_pq3t · 2022-10-16

**Confidence:** 4
**Correctness:** 4
**Technical Novelty And Significance:** 4
**Empirical Novelty And Significance:** 4
**Recommendation:** 10

**Clarity, Quality, Novelty And Reproducibility:**

Overall, I enjoyed reading this paper. The authors did a great job of motivating their approach, discussing relevant background, and describing the technique.

The technique itself and the experiments both appear to be high quality; the framework is principled and general while also turning into a fairly straightforward implementation. Sufficient details appear to be provided to reproduce the work.

The main thing I did not understand was whether the parameterization of $\mathbb{P}^{\theta}$ in (10) is indeed sufficiently rich to exactly achieve $\mathbb{P}^{\theta} = \mathbb{Q}^{\Pi^*}$. In particular, $\mathbb{P}^{\theta}$ appears to be Markov by design, whereas $\mathbb{Q}^{\Pi^*}$ is a mixture distribution and seems potentially non-Markov. Additionally (12) explicitly allows $\mathbb{Q}^{x}$ to be non-Markov. As such, it's not obvious to me that the model will always converge toward the correct solution.

My main questions thus are:
1. Is it the case that $\mathbb{Q}^{\Pi^*}$ is actually Markov for the suggested diffusion models, e.g. for those used in Algorithm 1?
2. When $\mathbb{Q}^{\Pi^*}$ is not Markov, does minimizing (13) still produce processes whose marginals are the same, e.g. that $\mathbb{P}^\theta_t = \mathbb{Q}^{\Pi^*}_t$ for all $t$ (and in particular for $t = T$)?

If 2 is true, that would be a very strong reason to prefer this training strategy in its full generality. If 1 is true, that would also be good motivation for the specific algorithm used, although it would be important to discuss the conditions under which it works (perhaps it has to do with $z_0$ being chosen as a single point rather than a distribution?). If neither are true, it's not obvious to me why it makes sense to minimize (13), other than "because it works well empirically". In any case I think this is an important discussion to include in the main paper.

There's some discussion of Markov-ness in the appendix, and Proposition A.3 seems to imply that the answer to my question 1 is yes, but I'm not sure. I was also confused by the claim "On the other hand, when $v_0 = 0$, we have that $Q^x = Q(\cdot|Z_T = x)$ ... and hence $\mathbb{Q}^{\Pi^*}$ is Markov following Proposition A.2."; shouldn't this be $Q^x = Q(\cdot|Z_0 = 0, Z_T = x)$, in which case Proposition A.2 wouldn't apply?

(A related but somewhat tangential question: Does minimizing (13) over the space of all functions always produce an $f^\theta$ that satisfies the condition in Proposition 2.3?)

In terms of novelty, using Doob's h-transform to enforce constraints as part of the parameterization of $\mathbb{P}^{\theta}$ seems quite novel and is a significant contribution. On the other hand, the definition of $\mathbb{Q}^{\Pi^*}$ seems to be essentially the same as the corruption process discussed by Peluchetti (2021) and Wu et al. (2022), since the constraint set doesn't play a role there; the authors do acknowledge this but I feel that the connection could be discussed a bit more. Two other pieces of related work which might also be worth including:
- [Li et al. (2022)](https://arxiv.org/abs/2205.14217) perform continuous-space diffusion for text using a rounding-based method
- [Bortoli et al. (2021)](https://arxiv.org/abs/2106.01357v1) give another diffusion process based on Schrodinger bridges, which seem related

A few other minor comments:
- Citations are oddly formatted at the end of section 2 in "referred to (e.g. Oksendal, 2013 ...) for more background"; perhaps this should be `\citet` instead of `\citep`?
- I was a bit confused by equation (6), since $z$ is used on the RHS but only $dz$ is present on the LHS. Is this because $dz$ is assumed so small that every $z$ in it has the same value of $q_{T|0}(x|z)$? (Perhaps this could be clarified in a footnote, and/or $z \in dz$ could be included somewhere in the equation?)
- In the equation between (8) and (9), should $Z_T = dx$ be $Z_T \in dx$?
- In section 2.3 at the end of page 4: I'm not sure it's true that most neural networks have bounded outputs; this seems very dependent on the architecture, although it would be very easy to enforce in practice.
- What does "in a way that is made precise in sequel" refer to? Is it just the second half of the sentence (in which case this seems unnecessary) or is it supposed to refer to something else?

**Strength And Weaknesses:**

Strengths:

- The paper is very clearly written, and provides sufficient background to understand the approach for readers who are unfamiliar with bridge processes and Doob's h-transform.
- The technique itself is very elegant and principled, and appears to be both simple to implement and widely applicable.
- The authors provide both a general, high-level framework (for arbitrary noise processes and constraint sets) and a concrete realization of it (using simple Brownian motion and constraints consisting of discrete sets, bounded intervals, and product spaces) which could be readily implemented for new tasks.
- Empirically, the results are very strong across a wide range of existing tasks, outperforming many individual models designed for specific tasks.

Weaknesses:

- **(addressed by new revision)** It's not obvious to me that the learned process $\mathbb{P}^{\theta}$ can exactly capture the fixed posterior $\mathbb{Q}^{\Pi^*}$ used for training, and thus that minimizing the loss in (13) is sufficient in theory to obtain a correct terminal distribution. (But perhaps I am not understanding something. Empirically, at least, it seems to be able to do this in practice.)
- The authors could discuss connections to prior work in a bit more depth.
- Since the trained model and inference process use discrete step sizes in practice, the learned model doesn't always generate values within the constraint set, so some form of rounding is still necessary. (But hopefully the amount of rounding required is quite small!)

**Summary Of The Paper:**

This paper proposes a new technique for learning generative diffusion-based models of data in constrained (discrete / bounded / mixed) domains. Interestingly, the diffusion process occurs in continuous time and space, but is "pinned" to always generate outputs in a potentially discrete constrained set; this pinning is accomplished by incorporating Doob's h-transform into the learned process.

The paper starts with a clearly written introduction to x-bridges (constrained to points), the h-transform, and $\Omega$-bridges (constrained to sets), and then introduces a parameterized family of such bridges, which can be used as a generative model. Next, it describes an approach for learning these bridges by minimizing the KL divergence between the parameterized family and a ground-truth mixture of point bridges, which turns out to be a straightforward score-matching objective.

Finally, the authors present results on a wide variety of constrained domains introduced by previous works, including mixed-categorical-continuous tabular data, point clouds, categorical segmentation maps, and discretized images. Across all domains and metrics, their approach achieves improved results.

**Summary Of The Review:**

Overall this is a good paper, which clearly describes relevant background, introduces a novel and elegant technique, and shows that it does very well across a large number of tasks of interest to the community. As such, I think this paper should be accepted. I think this paper could be even stronger with a discussion of the conditions under which minimizing the proposed loss will guarantee that the learned model faithfully represents the data distribution given enough capacity (or, even better, a proof that this is always the case).

*Updated review:* The authors have added a significantly expanded derivation in the appendix, which does indeed show that minimizing the proposed objective under the proposed parameterization exactly recovers the data distribution (and is furthermore a Markovization of the constrained diffusion process). Since this method appears to be both theoretically principled and empirically very strong, I have increased my score from 8 to 10. (However, most of the theoretical justification is in the appendix, and might be easy for readers to miss; I hope the authors can emphasize their theoretical contributions more in the final revision if this work is accepted.)

---

> ### Author Response · Authors · 2022-11-18
> **Response to Reviewer pq3t**
>
> Thank you for your positive rating!
>
> **Q1: Rounding**
>
> Yes, in practice we apply rounding due to the numerical error. Please refer to Appendix B. In Figure 3 of the main text and Figure 8 of Appendix B.4, we show the histogram of generated values before rounding, and as you said, the rounding error is small.
>
> **Q2: Markovian and Correctness of Eq. (13)**
>
> The answer to your question 2 is **YES**. As we show in the revised Appendix A.3,  $Q^{\Pi^*}$ is not necessarily Markov,
> and the learned $P^{\theta^*}$ is actually a *Markovization* of $Q^{\Pi^*}$, which is the Markov model that has the minimum KL divergence with $Q^{\Pi^*}$. A key property of Markovization is that we have the same  marginals $P^{\theta^*}_t = Q^{\Pi^*}_t$ at all time $t$, even though the joint distribution $P^{\theta^*}$ and $Q^{\Pi^*}$ are different.
>
> **Q3: I was also confused by the claim "On the other hand, when $v_0=0$, we have that $Q^x = Q(\cdot|Z_t=x)\dots$"**
>
> Here we want to discuss the condition when $Q^{\Pi^*}$ is/isn't Markov, as it would decide if $P^{\theta^*}$ is identical to $Q^{\Pi^*}$, or if it is just its Markovization. What Proposition A.3 (changed to Prop A.7 in the revision) is that, when $Q^x$ starts from a deterministic initialization $Z_0 = z_0$ (with zero variance $v_0 = 0$), then  $Q^x$ is the Brownian bridge starting from $Z_0 = 0$ and ending at $Z_T = x$.
>
> **Q4: Does minimizing (13) over the space of all functions always produce an $f_\theta$ that satisfies the condition in Proposition 2.3?**
>
> If the optimization in (13) is exactly solved, then the learned $P^{\theta^*}$ should equal to the Markovization of $Q^{\Pi^*}$ that we mentioned above. In this case, because $P^{\theta^*}$ and $Q^{\Pi^*}$ share the same marginal distributions at all time $t$, $P^{\theta^*}$ must be an $\Omega$-bridge, as $Q^{\Pi^*}$ is so, and there is no need to use Proposition 2.3.
>
> **Q5: I was a bit confused by equation (6)**
>
> This is a notation convention in measure theory and calculus. $d z$ can be formally viewed as an  infinitesimal volume centering around $z$. Hence, Eq 6 says that the infinitesimal measure $Q_{0|T}(d z | x)$ equals the $Q(dz )$ weighted by the conditioned density $q_{0|T}(x|z)$.  Hence, there is no need to have $dz $ in $q_{0|T}(x|z)$. We have clarified this notation issue in the revision.
>
> **Q6: I'm not sure it's true that most neural networks have bounded outputs**
>
> Please refer to General Response #2.
>
> **Q7: What does 'in sequel' refer to?**
>
> Thank you. We have revised our manuscript and explicitly refers `in sequel' to Appendix A.
>
> **Q8 : Minor: Typos, Citations, etc.**
>
> Thank you! We have added citations and fixed the typos in the revision. We will provide a more in-depth discussion of these related works in the final revision given more page limits.

---

> > ### Comment · Reviewer_pq3t · 2022-11-19
> > **Updated review and follow-up comments**
> >
> > Thank you for clarifying the relationship between $\mathbb{P}^\theta$ and $\mathbb{Q}^{\Pi^*}$ in the expanded Appendix A. To summarize, it looks like the answers to both my questions are yes:
> >
> > 1. When starting from a deterministic initialization, the $\mathbb{Q}^{\Pi^*}$ process is exactly Markov, so a sufficiently expressive model could exactly recover $\mathbb{P}^\theta = \mathbb{Q}^{\Pi^*}$ by minimizing the KL.
> > 2. In other cases, $\mathbb{Q}^{\Pi^*}$ may not be Markov, so the KL may not reach zero, but minimizing the KL (for a sufficiently expressive model) nevertheless leads to $\mathbb{P}^\theta$ recovering all of the marginals of $\mathbb{Q}^{\Pi^*}$ and in particular leads to recovering the desired generative distribution $\mathbb{Q}^{\Pi^*}_T$.
> >
> > I have increased my rating to 10, as the results in the appendix seems quite interesting and are a strong argument in favor of using this type of parameterization for generative models on constrained domains. However, I am surprised that this contribution is not given much emphasis in the main paper, which just states
> >
> > > $\mathbb{P}^\theta$ can approximate the given $\mathbb{Q}^{\Pi^*}_T$ well enough (in a way that is made
> > > precise in the Appendix A) such that their terminal distributions are close: $\mathbb{P}^\theta \approx \mathbb{Q}^{\Pi^*}_T = \Pi^*$.
> >
> > I think many readers may not understand from this that $\mathbb{P}^\theta$ is *provably equal* to $\Pi^*$ when the KL divergence attains its minimum under the parameterization in (10), even if that KL divergence does not reach zero due to the constraint. I would encourage the authors to actually state Proposition A.2 here instead of hiding it in the appendix; personally, I find Proposition A.2 to be a much stronger justification for using this approach than the high-level argument about EM that occupies most of Section 3. (Unfortunately I belive the period for uploading new revisions during the review period is now closed, but I hope that this change can be made in the final revision if this work is accepted.)
> >
> > A few additional comments:
> > - Regarding Q3: Thanks for the clarification. If I understand correctly, is the idea that the original process $Q$ is taken to be initialized from $Z_0 = 0$, which is still Markov, and then conditioning on $Z_T = x$ preserves this Markov-ness according to Proposition A.7?
> > - Regarding Q6 / General Response 2: It makes sense that the condition in Prop. 2.3 would be fairly mild even when the neural network is not bounded. However, the paper still seems to assert that most neural networks are bounded everywhere, which doesn't seem obvious. Since this isn't really necessary for the claim in Prop. 2.3 I'd suggest removing it or substituting a weaker claim (perhaps you could say that it's easy to enforce boundedness by clipping if necessary).
> > - Two more typos I noticed:
> >   - In Example 2.2 are $\mu$ and $z$ supposed to be the same? Otherwise I'm not sure what $\mu$ is.
> >   - Definition A.9: "reciporcal"

---

> > > ### Author Response · Authors · 2022-11-20
> > > **really appreciate the careful and expert review!**
> > >
> > > Dear Review,
> > >
> > > First, let us first thank you for your careful and professional review. We are really glad that you appreciate the theoretical value of work. Your understanding on the Markovization is perfectly correct.
> > >
> > > In fact, the results on Markovization was a main part of the draft in a previous submission of the draft. At the time, all the reviewers are not familiar with the theory, and rejected the paper based on complains about the theoretical complication. To avoid a similar situation, we rewrote the draft in more practical-oriented and accessible way and dropped the theoretical results that are not directly related to practical algorithms, which unfortunately disrupted the original presentation flow and caused the confusions that the reviewer correctly observed. But given that we have done this before, we think we can introduce the Markovization result back to the main paper in a clear way in the revision when an extra page is available.
> > >
> > > We truly appreciate that the reviewer identified this problem and gave us a chance/motivation to clarity and elaborate this issue. To summarize, the "EM" framework used in diffusion models differs from standard EM in two ways: 1) the drop of the E-step by using a simple $\mathbb Q^x$ surrogate, and 2) the issue of Markovization (which is an interesting model misspecification that does not actually cause bias in the final learning result). We think the drop of E-step is know to many but the Markovization issue is our theoretical contribution.

---

> > > > ### Author Response · Authors · 2022-11-20
> > > > **more**
> > > >
> > > > Regarding Q3: Yes, you are correct. If $\mathbb Q$ is initialized from a non-deterministic point $\mathbb Q_0$, then the initialization of the conditioned process $\mathbb Q^x = \mathbb Q(\cdot |Z_T = x )$, denoted by $\mathbb Q_0^{x}$, would depend on $x$ via Bayes rule: $\mathbb Q_0^{x}(d z_0) \propto \mathbb Q_0(d z_0) * q_{T|0}(x | z_0)$, where $q_{T|0}(x | z_0)$ is the transition density from $Z_0 = z_0$ to $Z_T = x$ under $\mathbb Q$. So the main problem is that $\mathbb Q_0^{x}$ depends on the final point $x$, except the special case when $\mathbb Q^0$ is a delta measure when multiplying with $p_{T|0}(x | z_0)$ of different $x$ yields no difference on $\mathbb Q^x_0$. This is why deterministic initialization is special here.
> > > >
> > > > Regarding Q6: We will replace the boundness condition with the moment-based argument in the general response, with possibility of using clipping for boundless as an argument.
> > > >
> > > > Thanks for pointing out the typos. We will fix them carefully.

---

### Official Review · Reviewer_WZ2o · 2022-10-22

**Confidence:** 2
**Correctness:** 2
**Technical Novelty And Significance:** 4
**Empirical Novelty And Significance:** 4
**Recommendation:** 6

**Clarity, Quality, Novelty And Reproducibility:**


**Clarity**
1. I am not convinced why Doob's h transform can be extended from point to distribution;
2. are we using two networks or one network? it seems that $f^{\theta}$ is one network, however, in section 3, the intractable $\mathbb{P}^{\theta, x}$ (due to the presence of neural force field; later replaced by $\mathbb{Q}^x$) also seems to be approximated somehow. It seems to me that $\eta^x$ alone is not trivial to estimate, not to mention about $\eta^{\Omega}$ which includes all possible data points $x$.


**Strength And Weaknesses:**

**Pros:**

1. The use of Doob's h-transform to steer the particles toward the desired point is an interesting idea and seems promising.


**Cons:**

Several claims may need to be justified:

1. My **biggest concern** is if Doob's h transform can be extended to model the data distribution: since Doob's h transform is often used to steer the solution to some point at the terminal time, while the authors seem to naturally extend that to distributions. I suspect that this may **not be a trivial extension**. Any reference supporting similar ideas would be greatly appreciated.

Minor:

1. The use of the path measure $\mathbb{Q}$ is slightly confusing. $d\mathbb{Q}_t/dx$ is often referred to as a distribution instead of $\mathbb{Q}_t$.

2. I don't understand what $Z_T\in dx$ means, it seems weird.

Other minor issues:
3. The figures and table in the appendix should use a better format. (larger font / y axis)

**Summary Of The Paper:**

The authors proposed to leverage Doob's h-transform to learn diffusion models within a constraint domain, including product spaces of any type, such as discrete, categorical, and their mix. Various experiments are conducted to demonstrate the interesting applications of such an algorithm.

**Summary Of The Review:**

I like the idea of Doob's h transform in studying diffusion models in a constrained domain, however, a few fundamental claims seem to be quite rushed and not fully verified. There are quite a few weird mathematical notations with kind of ambitious claims. I would suggest the authors elaborate more on the methodology part and why Doob's h transform extends from point to distribution.


========= post-review ==============

The variational gap should not be a severe issue due to the novelty in the methodologies to extend score matching with other (e.g. discrete) state spaces.

---

> ### Author Response · Authors · 2022-11-18
> **Response to Reviewer WZ2o**
>
> Thank you for the questions! We look forward to an increased rating if your concerns are properly addressed.
>
> **Q1: My biggest concern is if Doob's h transform can be extended to model the data distribution.**
>
> The case of conditioned process with a fixed terminal point is a standard form of $h$-transform.
> See Theorem 7.11 on Page 115 of https://users.aalto.fi/~asolin/sde-book/sde-book.pdf.
>
> $Q^x$ has a fixed terminal point, which is $x$, therefore $h$-transform can be applied. Then, by composing all the $Q^x$ together, we get $Q^{\Pi^*}$ whose end point is randomly drawn from the data distribution $\Pi^*$. We suggest the reviewer to read Section 3 (especially around Eq. (11)) again to understand how $Q^{\Pi^*}$ is constructed. For more rigorous analysis, we refer the reviewer to Appendix A, which we have revised to elaborate the theoretical foundation of the learning process in Section 3.
>
> **Q2: The use of the path measure $Q$ is slightly confusing. $d Q / d x$  is often referred to as a distribution instead of $Q_t$.**
>
> We have clarified the notation in the revision.
> $dz$ denotes a infinitesimal volume around $z$, and $Q(dz) = Q(x\in dz)$ denotes the probability measure of the infinitesimal set $dz$. Hence, $Q(dz)/dz$ can also be used to denote the density at point $z$.
>
> **Q3: 'Are we using two networks or one network?' and more**
>
> We are only using **ONE** neural network, which is $f_\theta$. The intractable $P^{\theta,x}$ is never needed because
> we simply replaced it with a simple and tractable $Q^x$ as a surrogate,
> thanks to the overparamterization property
> as we explained in the paragraph under Eq (11).
> Having a separate network to estimate $P^{\theta,x}$ would yield a typical EM, or VAE style algorithm, which is exactly we want we tried to avoid.
>
> Both $\eta^x$ and $\eta^{\Omega}$ have simple closed form when we derive them using $h$-transform with $Q$ set to be an elementary process and $\Omega$ has a factorized form. For example, when $Q$ is a time-scaled Brownian motion, we simply have $\eta^x = \frac{z-x}{\beta_T-\beta_t}$
> and $\eta^{\Omega}$ can be calculated easily with the procedure we show in General Response \#1.

---

> > ### Comment · Reviewer_WZ2o · 2022-11-23
> > **Is section 3 learning the ELBO?**
> >
> > Is section 3 essentially learning the ELBO (in terms of path measures), which provides a computationally efficient way to solve the $\Omega$-bridge, but it doesn't guarantee the generated distribution $\mathbb{Q}_T$ at time $T$ to match the target data distribution $\Pi^{*}$ due to the limitation of the EM algorithm?
> >
> >
> > Extra (but not required): It would be convincing to demonstrate the sample quality via a tiny 1-D (mixture) distribution to see how far away $\mathbb{Q}_T$ deviates from $\Pi^*$ (a clean code for a simple baby example is more appealing to me than these cool experiments).

---

> > > ### Author Response · Authors · 2022-11-24
> > > **further clarification**
> > >
> > > Thanks a lot for your question. We think you still have some misunderstanding on the method as we  elaborate below. We would appreciate any follow up discussion if the comments below don't address your question.
> > >
> > > 1) $\mathbb Q_T$, which is the $T$-marginal distribution of the reference process $\mathbb Q$ is *irrelevant* to the algorithm, as the algorithm only depends on the conditioned process $\mathbb Q^x = \mathbb Q(\cdot | X_T = x )$, which is independent of $\mathbb Q_T$.
> > >
> > > 2) $\mathbb Q_T^{\Pi^*}$, which is the $T$-marginal distribution of $\mathbb Q^{\Pi^*} = \int Q^x(\cdot) \Pi^*(d x)$, is guaranteed to  equal to $\Pi^*$ by definition, because $\mathbb Q^{\Pi^*}$ is the mixture of $Q^x$ with $x$ drawn from $\Pi^*$.
> > >
> > > (Due to above, there is no need to compare either $\mathbb Q_T$ or $\mathbb Q_T^{\Pi^*}$ with $\Pi^*$ on toy.)
> > >
> > > 3) A traditional maximum likelihood of the problem would amount to minimize the marginal KL divergence: $L_T(\theta) = KL(\Pi^*|| \mathbb P^{\theta})$. It is easy to show that loss function  $\mathcal L(\theta) = KL(\mathbb Q^{\Pi^*}||\mathbb P^\theta)$ in Eq (11) is an upper bound in that $\mathcal L(\theta) \geq L_T(\theta)$ by the chain rule of KL divergence. In this sense Eq (11) can be viewed as a form of ELBO. However, simply viewing our method as an vanilla ELBO is not sufficient in explaining/understanding the algorithm, largely due to the special overparameterized nature that we discussed in Section 3:
> > >
> > > i) Because there are infinite many processes $\mathbb P^\theta$ that has the same final distribution $\mathbb P^\theta_T$,  the marginal KL divergence  $L_T(\theta)$ (and hence MLE) does not place any preference on which $\mathbb P^\theta$ should be selected, and hence yields an under-determined estimation problem, whose solution depends on training initialization and other hyper parameters in a unpredictable way. Hence $L_T(\theta)$ *is not* an idea objective to optimize, and there is no need to justify another objective function (such as $\mathcal L(\theta)$) by showing that it is an upper or lower bound of $L_T(\theta)$.
> > >
> > > ii) As we show in Section 3, again due to the special over-parameterized nature of the problem, the exact minimum of $\mathcal L(\theta)$ recovers the true data distribution, even when the surrogate $\mathbb Q^x$ does not equal the exact conditioned process $\mathbb P^{\theta,x} = \mathbb P^{\theta}(\cdot | X_T = x)$. In this case, $\mathcal L(\theta) > L_T(\theta)$ serves as a strict upper bound, except the case of true model $P^{\theta^*}$ when $\mathcal L(\theta^*) = L_T(\theta^*) = 0$.
> > >
> > > iii)  As we show in the discussion with  Reviewer pq3t and in Appendix A.3, when $\mathbb Q^{\Pi^*}$ is not a Markov process, the minimum $\mathcal P^{\theta^*}$ of  $KL(\mathbb Q^{\Pi^*}||\mathbb P^\theta)$ does not equal to $\mathbb Q^{\Pi^*}$ and instead equals its Markovization, even though it aways guarantees the matched marginal distribution $\mathbb P^{\theta^*} = \mathbb Q^{\Pi^*} = \Pi^*$. This is a special property that does not appear in typical (variational) EM and ELBO frameworks.

---

> > > > ### Comment · Reviewer_WZ2o · 2022-11-24
> > > > **my concern has been partially addressed.**
> > > >
> > > > Thanks for the response, my concern has been partially addressed. I have slightly increased my score (reluctantly) but reduced my confidence to the minimum due to the following reasons:
> > > >
> > > > Although fancy/ cool DNN experiments are appealing, **the lack of verification in a simple/ baby (but general) example** made me concerned about the **true approximation ability** (because many **wrong** algorithms can get decent results in DNN); this paper is more like a methodology paper (due to the rich experiments + theory support), so to demonstrate if a method works or not, a simpler dataset is also needed because it is easier to analyze; the lack of analysis on a sample approximation accuracy may limit this paper to become more scientific. Otherwise, it is just an extension of Doob's-h transform [1] to mixture cases.
> > > >
> > > >
> > > > [1] Simulating Diffusion Bridges with Score Matching.

---

> > > > > ### Author Response · Authors · 2022-11-29
> > > > > **Adding Toy Example and Hope to Fully Address Concerns**
> > > > >
> > > > > Dear Reviewer,
> > > > >
> > > > > Dear Reviewer,
> > > > >
> > > > > We show a toy example in the figure linked below. It shows clearly that our method can correctly recover the ground truth distribution. We did not thought about adding the toy because we thought the correctness of the algorithm is self-evidence from the theoretical derivation; now we agree that we should add a toy and give a careful discussion (even if it goes to appendix). If the reviewer still have any hesitation on the correctness of the method, please do let us know and we will try our best to address.
> > > > >
> > > > > https://anonymous.4open.science/r/ICLR2023_rebuttal_constrained_diffusion-B25F/rebuttal.png
> > > > >
> > > > > According to the reviewer’s requirement, we conduct a toy experiment on a 1D discrete distribution, where the value belongs to $\{-1, 0, 1\}$ and the probability are $0.15, 0.55, 0.3$ respectively. We set the initial distribution as a Delta function $Z_0=0$ and applies Integer-bridge based on $d Q = d W_t$ (which results in a standard Brownian bridge) as in Algorithm 1 and Experiment 4.4. We use a 3-layer MLP to parameterize $f^\theta$ and set the number of diffusion steps to $K=1000$.
> > > > >
> > > > > In Figure (a), we present the groundtruth distribution, which is $(0.15, 0.55, 0.3)$. In Figure (b), we show the empirical distribution sampled from our learned Integer-bridge with 10,000 samples, which is $(0.152, 0.544, 0.304)$. The empirical distribution matches the true distribution up to the numerical and fitting error of the neural network. In Figure (c), we show the diffusion trajectory of 10 random samples from the learned Integer-bridge. Driven by $\eta^\Omega$, all the trajectories converges to $\{-1, 0, 1\}$ when $t=T$.

---

> > > > > > ### Comment · Reviewer_WZ2o · 2022-11-29
> > > > > > **Thanks for the illustration.**
> > > > > >
> > > > > > To improve the impact of this paper, the authors may consider releasing your code (especially the baby example) in the final version for researchers to study.

---

> > > > > ### Author Response · Authors · 2022-12-02
> > > > > **Will release the code and addressing the error analysis concern**
> > > > >
> > > > > Dear reviewer,  Thanks a lot. It is a part of our plan to release the code once the paper is accepted.
> > > > >
> > > > > Moreover, below we want to address your last comment of *"the lack of analysis on a sample approximation accuracy may limit this paper to become more scientific"*.
> > > > >
> > > > >  - We did have a set of theoretical results on analyzing the approximation accuracy. See Appendix A.5 and Proposition A.18 (main result) in the anonymous link below:
> > > > >
> > > > > https://anonymous.4open.science/r/ICLR2023_rebuttal_constrained_diffusion-B25F/ICLR2023_revision.pdf
> > > > >
> > > > > It shows that, in the case when the domain $\Omega$ is $\mathbb R^d$, the KL divergence between the learned distribution and the true distribution is $O(\frac{\sqrt{\log(1/\epsilon)}}{n}  + \sqrt{\epsilon})$, where $\epsilon$ is the time-discretization step size of the diffusion process and $n$ is the number of data points. We will extend the result to more general $\Omega$ in future work.
> > > > >
> > > > > We think, in general, that our method is a novel and theoretically principled approach for learning generative models on different domains.
> > > > >
> > > > > - We also want to point out that it is common for the first methodology paper to focus on proposing the main algorithm idea, and leave the refined the theoretical analysis to future works (none of the original score-based and denoting diffusion papers have error analysis. In fact, the paper [1] (Simulating Diffusion Bridges with Score Matching) cited by the reviewer also did not have an error analysis).
> > > > >
> > > > > - Moreover, the full course discretization error analysis of our method does  require a lot of technical works (the $\Omega=\mathbb R^d$ already required a number of pages of proofs as we show in the link), and it does not influence the validity of the main method, which we think is self-evident from the derivation we have. Hence we think it will require a series of works in multiple years to fully address the gap. It is certainly something we should expect in a 9-page ICLR paper, especially given that we already have an abundant of empirical and theoretical results.
> > > > >
> > > > > We would appreciate the reviewer to further raise the score if our argument is reasonable. Otherwise, we are happy to answer further questions to *fully address your concerns.*

---

### Official Review · Reviewer_bJKz · 2022-10-23

**Confidence:** 4
**Correctness:** 4
**Technical Novelty And Significance:** 3
**Empirical Novelty And Significance:** 3
**Recommendation:** 8

**Clarity, Quality, Novelty And Reproducibility:**

Clarity
-------
I found the paper to be reasonably clear except for (notably) the exposition. In particular, the exposition, while trying to remain faithful perhaps to the field of stoch analysis, greatly sacrifices clarity for generality. I would suggest the authors look into trimming this section into a more digestible form, perhaps moving general constructions to the appendix. There are also some minor typos that should be cleaned up (e.g. "an simpler" in page 1).

Quality
--------
The overall quality of the paper is reasonably high, as the methodology + experimental sections are reasonably extensive, and the idea is rather clean.

Novelty
---------
The paper is reasonably novel (introduces a new method based on existing literature in stoch analysis for an existing tasks). However, the comparison with concurrent work (Key et al. 2022 in the paper) should be extended with more details.

Reproducibility
-----------------
The reproducibility is quite poor. No code is included (crucial for some experiments), and the appendix doesn't have key architectural details that are nontrivial (e.g. which activations are used and do they make the neural network bounded as per requirement). Additionally, the training scripts for the ML methods used in Sec 4.1 need to be included as well (since accuracy of these models is a highly brittle metric).

**Strength And Weaknesses:**

Strengths
-----------
* The method is theoretically-principled (building off of known results in the stoch analysis literature). In addition, it is reasonably clean (for example avoiding simulating an SDE as is usual for diffusion models).
* The empirical results are very convincing. In particular, there is extensive evaluation on many different datasets, including some applications of discrete diffusion which are novel.
* The idea seems impactful. In particular, there seems to be a lot of work on learning diffusion models which generate discrete structured data.

Weaknesses
---------------
* The proposed claim about diffusion processes that end on general compact spaces is not really supported. In particular, note that the proposed methodology would require an intractable integral over the whole domain. This is supported by the fact that experiments don't showcase this, even though such examples exist in the literature [1]. I would remove this claim especially in light of existing literature for diffusion models on manifolds.
* There seems to be several technical limitations to the proposed approach:

1) Since the gradient/eta term goes to infinity for $x \notin \Omega$,  there should be some numerical difficulties when training and sampling. In particular, since $f$ is constructed to be bounded, any small amount of numerical error should cause significant problems. Furthermore, for sampling, I imagine that really small step size must be used (or it should be annealed as $t \to T$).

2) Prop 2.3 is presented as a simple requirement that can be easily overcome (by making the neural network bounded). However, this only works when the activations are bounded (assuming an MLP like structure) and are not something like ReLU. Furthermore, while it is true that we still retain universal approximation, how does this work with the numerical blowup of the gradient/eta term (as described in 1)? In particular, does this significantly hamper training?

3) The evaluation of the gradient/eta term requires a discrete summation over all points in the domain. This seems like it would very computational expensive (akin to the computational blowup of Gaussian Mixture Models). For experiments in this paper, this amounts to evaluating a 256^{32 x 32 x 3} GMM (at least for CIFAR). I don't believe that the paper touches upon this, and I also don't think naive techniques like MC sampling would work during training/sampling (due to pole-nature of the sampled points).

4) The paper does not present a way to calculate exact log-likelihoods. This may be difficult since we are projecting from a continuous space to a discrete space, but this still represents a significant limitation when compared to vanilla SDE-diffusion models which normally have this.

* The (Key et al. 2022) paper that is mentioned seems very similar, and the paper should devote more time and space to explain the differences. (It is technically concurrent work since it appeared in arxiv less than a month before ICLR submissions).
* Some of the metrics used for evaluating the datasets are quite weak. For example, the experiments in 4.1 train another ML model to fit the generated datapoints, which seems like a very difficult comparison to make since there are so many hyperparameters to tune.

[1] https://arxiv.org/abs/2202.02763

**Summary Of The Paper:**

This paper proposes a method to learn diffusion models which is forced to converge to a specified set of points. It does this by using Doob's h-transform, a classical tool in stochastic calculus. Experimental results show a general improvement when modeling data that has some sort of discrete structure (e.g. image pixel values).

**Summary Of The Review:**

Overall, I lean to accept the paper as the method is relatively nice and the results are very convincing. Importantly, I found there to be some problems in the methodological setup, but this is to be expected due to the nature of the problem (continuous to discrete). My rating is conditioned on the authors responding to the comments above and updating the manuscript to reflect the limitations.

---

> ### Author Response · Authors · 2022-11-18
> **Response to Reviewer bJKz**
>
> Thank you for your positive rating!
>
> **Q1: The proposed claim about diffusion processes that end on general compact spaces is not really supported.**
>
> We did not intend to claim that our method works for general compact spaces. Rather, as we said in the intro (bottom paragraph of page 1),  our method applies to any domains on which a properly defined summation/integration (for calculating $\eta^\Omega$) is tractable. This includes a large class of domains that are product of one dimensional sets.
>
> Please let us know which part of paper that gives the impression of over-claiming. We will fix it right away.
>
> Thanks for pointing out Riemannian SGM. We have already added citation in the revised version. We will clarify the differences/connections with Riemannian SGM given more page limits in the future.
>
> **Q2: Since the gradient/eta term goes to infinity for $x \not \in \Omega$ , there should be some numerical difficulties when training and sampling.**
>
> Even though $\eta^\Omega$ goes to infinity as $t\to T$, we found it surprisingly stable w.r.t. numerical discretization. Perhaps the easiest way to see this is through the Brownian bridge $d Z_t = \frac{x-Z_t}{T-t} dt + dW_t$.
> Assume we use step size $\epsilon = 1/N$ for $N$ steps. Then in the last step, we have $t = \frac{N-1}{N}$, and hence $T-t=1/N$. Therefore, we have
> $$
> \hat Z_{1} = \hat Z_{(N-1)/N}  + \epsilon \frac{x-\hat Z_{(N-1)/N}}{T-t} + \sqrt{\epsilon} \xi  =
> x + \sqrt{\epsilon} \xi,$$
> where $\xi$ is standard Gaussian noise. This suggests that we have $||\hat Z_1 - x||^2 = O(1/N)$.
> Intuitively, we think the stability comes from the fact that $\eta^\Omega$ always point to the same set $\Omega$, which prevents the accumulation of harmful error, and increasing magnitude only increases the speed of converging to $\Omega$.
>
> **Q3: Prop 2.3 is presented as a simple requirement that can be easily overcome (by making the neural network bounded). However, this only works when the activations are bounded (assuming an MLP like structure) and are not something like ReLU.**
>
> Please refer to General Response #2.
>
> **Q4: The evaluation of the gradient/eta term requires a discrete summation over all points in the domain. This seems like it would very computational expensive (akin to the computational blowup of Gaussian Mixture Models).**
>
> Please refer to General Response #1.
>
> **Q5: The paper does not present a way to calculate exact log-likelihoods.**
>
> Thank you for pointing this out. While SDE-diffusion models can calculate exact log-likelihood with ODE for continuous data, they still require techniques like uniform dequantization to deal with discrete data. It is known that log-likelihood of the continuous model on uniformly dequantized data is a lower bound of the corresponding discrete model. For discrete counterparts we mentioned in the paper, Argmax Flow and D3PM, they are also limited to variational bounds. We agree that how to compute the exact likelihood for discrete data is a very interesting problem and we leave that for future work.
>
> **Q6: Related Work: (Key et al. 2022)**
>
> We would like to discuss it if the reviewer could give us a link to the paper.
>
> **Q7: The metrics in the experiments in Section 4.1**
>
> In fact, how to measure the quality of generated data is itself a very difficult problem, especially for tabular data. We follow previous work [1], which is well-known in the field of tabular data generation, to measure machine learning efficacy as evaluation metric. In the experiment of Section 4.1, we adopted the evaluation scripts from the Synthetic Data Vault repository as stated in Appendix B, which is famous for generating single-table, multi-table and timeseries datasets. We simply use the default hyper-parameters for the evaluation without further tuning. To mitigate bias, we provide results of three different ML models, logistic regression, AdaBoost classifier and MLP classifier. We think this is fair enough to say that the data synthesized by our method is better.
>
> [1] Modeling Tabular Data using Conditional GAN
>
> **Q8: Reproducibility**
>
> We have provided the network structure, baseline configuration, data pre-processing strategy, metrics, etc., for each experiment in the experiment section and Appendix B. The training scripts for the metrics in Sec.4.1 are adopted from Synthetic Data Vault, as stated in Appendix B.1. Our method does not ask for special network structure as explained in General Response \#2. We have revised the manuscript according to the reviewer's requirement. We will open-source upon acceptance.
>
> **Q9: Limitations**
>
> We had to make the "Conclusion and Limitation" section really short due to the space limit. We did mention the need of systematic studies on various parameters choices, albeit very briefly. We will further elaborate on more limitations in the final version, such as the requirement of having $\Omega$ to have a factorized form. We have added a short sentence regarding this in the current revised version.

---

> > ### Comment · Reviewer_bJKz · 2022-11-23
> > **Thank you for your response.**
> >
> > I thank the authors for their response, as it has mostly addressed my main concerns/questions. Since my main concerns were with experimental results/runtime, I will be raising my score to an accept.
> >
> > Q1: For general infinite compact domains, the problem seems to be that evaluating the sum would be untractable (which is why I assume there are no experimental results here). I would still advise the authors to change their language (e.g. remove references to "general constrained domain" and add a disclaimer to the remark in algo 1).
> >
> > Q6: This was a typo. Sorry, this was supposed to be Ye et al 2022 as in "Ye et al. (2022), which proposes to learn first hitting diffusion models for generation on both discrete sets and spheres".

---

> > > ### Author Response · Authors · 2022-11-24
> > > **Thank you!**
> > >
> > > Thank you for raising the score. We are happy that your concerns are properly addressed.
> > >
> > > For Q1 and Q6, we will add disclaimer and more discussion of concurrent work (Ye et al.) in the future version. Currently we are not able to revise the manuscript on OpenReview.
> > >
> > > Best,
> > >
> > > Authors

---

### Official Review · Reviewer_EaLU · 2022-10-29

**Confidence:** 3
**Correctness:** 4
**Technical Novelty And Significance:** 2
**Empirical Novelty And Significance:** 2
**Recommendation:** 8

**Clarity, Quality, Novelty And Reproducibility:**

I found the work quite clear and including useful theoretical presentation and useful experiments. There are a few parameters that are unclear, it would be helpful to be a bit more specific on parameter settings given these differ from the relevant comparisons a bit, currently it is difficult to judge how large these differences are.

Small things:

an —> a bottom of page 1

It would be useful to note the noise decay schedule used in prior work for a direct comparison.

**Strength And Weaknesses:**

Strengths:

- A conceptually simple framework for enforcing discrete generation that is theoretically sound and works well experimentally
- Comprehensive experiments on a number of domains including new domains, tabular data with both discrete and continuous data, and standard domains (discrete Cityscapes and CIFAR10).

Weaknesses:

- The GMM framework means that the current implementation is efficiently computable on discrete domains or domains that can be efficiently integrated over.
- No mention of compute times. From my understanding generating CIFAR10 images requires a distance computation to ~2^17 points each of the 1000 diffusion steps? This seems like it would be quite slow for large domains.
- For the Cityscapes and cifar10 comparisons, the parameters used are slightly different than those used in previous work. While these discrepancies are small, the performance improvements are also rather small. Since the numbers are taken from the prior papers, it would be useful to use the same setup as the prior papers for direct comparison of the methods.
- The limitations of this work are not discussed in the limitations section.

**Summary Of The Paper:**

The authors propose a method of learning diffusion models on constrained domains. Specifically, the propose to add a term composed of the gradient of a gaussian mixture model with bandwidth $T-t$ to encourage diffusion towards a domain embedded in Euclidean space. This is tested in a variety of domains against relevant models.

**Summary Of The Review:**

I found this paper an enjoyable read. I would be curious as to the extra time it takes to evaluate for some of the larger categorical spaces. I would also suggest more specific parameter specifications, and more discussion of the limitations of the h-transform in terms of efficiency.

---

> ### Author Response · Authors · 2022-11-18
> **Response to Reviewer EaLU**
>
> Thank you for your positive rating!
>
> **Q1: No mention of compute times.**
>
> Because the domain is factorized,
> the computation cost grows linearly with the dimension, rather than exponentially.
> Please refer to General Response #1 for the detailed explanation of the computation of $\eta^\Omega$.
>
> **Q2: For the Cityscapes and cifar10 comparisons, the parameters used are slightly different than those used in previous work. / It would be useful to note the noise decay schedule used in prior work for a direct comparison.**
>
> We could not directly adopt the same parameters from the baselines because our framework has significant difference and is not directly comparable with the baseline methods. For example, the diffusion process of Argmax Flow / D3PM works on the discrete space while ours work on the continuous space. However, we tried our best to match the settings in the experiments, including the network structures, pre-processing and the training/validation datasets. Based on the same reason, we do not think we can perform strict comparison with the noise decay schedule used in the baselines. Our experimental configuration are listed in Appendix B in detail.
>
> Like the case of many other works, it is often impossible to get absolutely apple-to-apple comparison due to the difference of the different methods. But we think our method is sufficient to demonstrate the potential and power of our method, especially when considering the novelty in methodology.
>
> **Q3: The limitations of this work are not discussed in the limitations section.**
>
> We had to make the "conclusion and Limitation" section really short due to the space limit. We did mention the need of systematic studies on various parameters choices, albeit very briefly. We will further elaborate on more limitations in the final version, such as the requirement of having $\Omega$ to have a factorized form. We have added a short sentence regarding this in the current revised version.
>
> **Q4: Typos**
>
> Thank you! we have fixed it in the revision.

---

> > ### Comment · Reviewer_EaLU · 2022-12-12
> > **Thank you for your response and clarifications**
> >
> > I have updated my score from 6 --> 7 with the additional discussions of limitations and clarifications on computational complexity for images.

---

### Public Comment · ~James_Thornton1 · 2022-11-08
**Existing work using h-transform with diffusion models to construct diffusion bridges**

Thank you for this interesting work. I would like to bring your attention to highly relevant work **Simulating Diffusion Bridges with Score Matching** [1], which has not been discussed.

[1] proposes what appears to be the **same methodology -- using h-transform with diffusion models to create a diffusion bridge** which is at the core of this work, and which this work builds on. I would argue [1] is not concurrent given it was publicly available, with code, about one year ago (November 2021).

[1] **Simulating Diffusion Bridges with Score Matching** (November 2021)
Jeremy Heng, Valentin De Bortoli, Arnaud Doucet, James Thornton, https://arxiv.org/abs/2111.07243

---

> ### Author Response · Authors · 2022-11-20
> **related but orthogonal works**
>
> Thanks a lot for pointing out the paper. We will cite and discuss it in the revision.
>
> We do want to point out that we think that using h-transform to derive diffusion bridges is a part of the standard toolbox in stochastic calculus, even though it has not yet been well known in the machine learning community. The main goal and contribution of this paper is a new method for learning *discrete* valued generative models, for which h-transform happens to play an important role for both constructing 1) the learning objective via $x$-bridges and 2) the model family via $\Omega$-bridges. The paper [1] pointed out here, on the other hand, applies h-transform to solve a very different problem: numerical simulation of the conditioned process of a diffusion process with the terminal state pinned at a fixed final state. The main difficult in [1] is that the conditioned process does not have a closed form solution and hence need to be approximated in certain way. In our work, because we assume that baseline process $\mathbb Q$ is simple Brownian motion, the conditioned process $\mathbb Q^x = \mathbb Q(\cdot | Z_t = x)$ has a closed form solution and hence no approximation is needed. Our work should not be viewed as conflicting with other papers who use the same tools (mostly only for the $x$-bridge part for constructing learning objective).
>
> We think the exact connection of our paper and paper [1] is as follows: If we follow the exact EM framework to estimate $\mathbb P^{\theta,x} = \mathbb P^\theta(\cdot | Z_T = x)$, rather than replacing $\mathbb P^{\theta,x}$ with $\mathbb Q^x$ (see our Section 3), then the method in paper [1] can be used as the missing E-step (while our method is the M-step).
> With this in mind, knowing that EM is a coordinate descent on KL divergence, it is not surprising to see that both our paper and paper [1] involves minimizing some KL divergence (but with respect different variables), which would necessarily reduce to  a form of mean square score matching as it should be in the case of diffusion processes by Girsanov theorem.

---

### Comment · Area_Chair_GK3b · 2022-11-15
**Discussion/Rebuttal**

Dear Authors,

A response or rebuttal to the reviews would be highly appreciated.

Kind regards,
Your AC

---

### Author Response · Authors · 2022-11-18
**Greetings and Response to the Reviewers**

We thank the reviewers for the comments and questions! We address common concerns from the reviewers in General Response #1 and #2. After that, we address the individual concerns from each reviewer. We would appreciate any increase in rating if your concern is properly addressed.

Best regards,

Authors

---

### Author Response · Authors · 2022-11-18
**General Response #1: How to calculate $\eta^\Omega$ and the cost**

Our $\eta^\Omega$ is computed with the following equation:

$$
\eta^{\Omega}(z,t)
= \sigma_t^2 E_{x\sim \mathcal{N}_{\Omega}(z,\beta_T-\beta_t)} \left [ \frac{x - z}{\beta_T - \beta_t}\right]
$$

As we stated in Example 2.2 and Appendix A.6, it is tractable to
calculate $\eta^{\Omega}$ once we can evaluate the expectation of $\mathcal{N}_{\Omega}(z,\beta_T-\beta_t)$.
A general case is when the domain is a product of a number of  one (or low) dimensional sets, say $\Omega = I_1\times \cdots I_d$, in which case the expectation reduces to one or low dimensional Gaussian integrals.

Specifically, when $\Omega = I_1\times \cdots I_d$ and each $I_i$ is a one dimensional set, we have

$$
\eta^{\Omega}(z, t) =\left  [\eta^{I_1}(z_1, t), \eta^{I_2}(z_2, t), \dots, \eta^{I_d}(z_d, t)\right ]^\top,
$$
where the $i$-th element of $\eta^\Omega$ is $\eta^{I_i}$, the drift force of the $I_i$-bridge,
$$
\eta^{I_i}(z_i,t)
= \sigma_t^2 E_{x_i\sim \mathcal{N}_{I_i}(z_i,\beta_T-\beta_t)} \left [ \frac{x_i - z_i}{\beta_T - \beta_t} \right].
$$
$\eta^{I_i}(z,t)$ can be calculated easily as it only involves a one-dimensional integration. This allows us to decompose the calculation of $\eta^\Omega$ in **a dimension-wise manner**.

Particularly, when $\Omega$ is the discrete CIFAR10 domain, where the size of the image is $3\times 32 \times 32$ and the number of categories is $256$, the computation needed is $256 \times 3 \times 32 \times 32$, not $256^{3\times 32 \times 32}$, and the implementation can be easily done with a softmax-like operator. In this case, the practical computational time for computing $\eta^\Omega$ is negligible compared with the forward and backward time taken by the neural network $f_\theta$. On our Nvidia Titan XP GPU, it takes $\sim 0.0015s$ for computing $\eta^\Omega$, and it takes $\sim 0.026s$ for the forward pass of $f_\theta$ and $\sim 0.064s$ for backward. We want to point out that, the inference and back-propagation time of $f_\theta$, which in the CIFAR10 task is a U-shape convolutional neural network  with over $30$ million parameters, is polynomial to the input size with a coefficient much larger than $256$. Therefore, even when the image size increases, the relative time cost for $\eta^\Omega$ is still small compared with $f_\theta$.

---

### Author Response · Authors · 2022-11-18
**General Response #2: The Requirement on the boundness of $f^\theta$**

The condition in Proposition 2.3 is
bounded L2 norm $E_{Z\sim Q^\Omega}[\int_0^T ||f_\theta(Z_t,t) ||^2 d t ] <+\infty$, a simple sufficient condition of which is when $f^\theta$ is bounded everywhere.

However, the bounded L2 norm is in general a very mild condition and can easily hold even when $f_\theta$ is not bounded.
For example, assume that $f_\theta(x) \leq a ||x||^\beta + b$, which holds for ReLU network with $\beta = 1$,
we just need to require that the underlying process has bounded moment $E_{Z\sim Q^\Omega}[\int_0^T ||Z_t||^{2\beta} d t] < +\infty$, which is a typical regularity condition to expect, especially considering the end point $Z_T$ is contained inside $\Omega$ which could be bounded.

---

### Decision · Program_Chairs · 2023-01-20

**Decision:**

Accept: notable-top-25%

**Justification For Why Not Higher Score:**

The experimental results section is sufficient, but not outstanding.

**Justification For Why Not Lower Score:**

See strengths of the paper in the meta-review above.

**Metareview: Summary, Strengths And Weaknesses:**

Ratings: 7/8/6/10
Confidence: 3/4/2/4

The authors propose a method of learning diffusion models on constrained domains. Specifically, the propose to add a term composed of the gradient of a gaussian mixture model to encourage diffusion towards a domain embedded in Euclidean space. This is tested in a variety of domains against relevant models.

The reviewers agree that the paper is well-written and well-motivated, and that the method is theoretically principled. In addition, the empirical results are convincing, with some novel applications of discrete diffusion. A small weakness is that the method only works if the discrete domain is not too large / is integrable.

After discussion and an updated draft, Reviewer pq3t changed his score from 8 to 10. The main concerns, but not all, of Reviewer EaLU were addressed in an author response. Likewise, Reviewer bJKz had a number of concerns, which are mostly addressed by the author response, such that the reviewer raised his score to accept. Reviewer WZ2o's concerns were all addressed by the authors. Reviewer pq3t is extremely positive, and had some minor concerns, which were mostly addressed by the author response.

A commenter pointed out the lack of comparison to related work "Simulating Diffusion Bridges with Score Matching" (November 2021). The authors addressed this in detail in a comment, and promised to update the draft (which he currently is unable to do).

**Note From Pc:**

if the above contains the word "oral" or "spotlight" please see: "oral" presentation means -> notable-top-5% and "spotlight" means -> notable-top-25%. As stated in our emails, we are disassociating presentation type from AC recommendations